# Structural insights into the BRAF monomer-to-dimer transition mediated by RAS binding

Juliana A. Martinez Fiesco[1,3], David E. Durrant [2,3], Deborah K. Morrison [2✉] & Ping Zhang [1✉]

RAF kinases are essential effectors of RAS, but how RAS binding initiates the conformational changes needed for autoinhibited RAF monomers to form active dimers has remained unclear. Here, we present cryo-electron microscopy structures of full-length BRAF complexes derived from mammalian cells: autoinhibited, monomeric BRAF:14-3-3$_2$:MEK and BRAF:14-3-3$_2$ complexes, and an inhibitor-bound, dimeric BRAF$_2$:14-3-3$_2$ complex, at 3.7, 4.1, and 3.9 Å resolution, respectively. In both autoinhibited, monomeric structures, the RAS binding domain (RBD) of BRAF is resolved, revealing that the RBD forms an extensive contact interface with the 14-3-3 protomer bound to the BRAF C-terminal site and that key basic residues required for RBD-RAS binding are exposed. Moreover, through structure-guided mutational studies, our findings indicate that RAS-RAF binding is a dynamic process and that RBD residues at the center of the RBD:14-3-3 interface have a dual function, first contributing to RAF auto-inhibition and then to the full spectrum of RAS-RBD interactions.

[1] Center for Structural Biology, Center for Cancer Research, National Cancer Institute-Frederick, Frederick, MD 21702, USA. [2] Laboratory of Cell and Developmental Signaling, Center for Cancer Research, National Cancer Institute-Frederick, Frederick, MD 21702, USA. [3] These authors contributed equally: Juliana A. Martinez Fiesco, David E. Durrant. ✉email: morrisod@mail.nih.gov; ping.zhang@nih.gov

The RAF kinases (ARAF, BRAF, and CRAF/RAF1) are key intermediates in the RAS pathway, functioning in the transmission of signals that regulate cell proliferation, differentiation, and survival[1]. Over the years, biochemical studies have provided critical insights regarding the mechanisms that regulate RAF signal output[2,3]. In quiescent cells, members of the RAF kinase family localize to the cytosol as inactive monomers[4], maintained in an autoinhibited state through intramolecular interactions between the RAF regulatory and catalytic domains and by the binding of a 14-3-3 dimer to two phosphorylation-dependent serine sites (pS365 and pS729 in BRAF)[5–8]. In response to signaling events and RAS activation, RAF interacts directly with GTP-bound RAS at the plasma membrane[9–12], which ultimately disrupts the autoinhibited state and promotes RAF dimerization and kinase activation[13–16].

Numerous structures of dimerized RAF kinase domains and other isolated regions of RAF have been solved using X-ray crystallography or NMR spectroscopy; however, the structural determination of full-length monomeric and dimeric RAF complexes has been challenging due to the number of proteins that associate with RAF as well as the largely unstructured nature of the N-terminal regulatory domain. Nevertheless, recent cryo-electron microscopy (cryo-EM) and crystal structures of BRAF have shed light on these complexes, providing visual context to the knowledge obtained through biochemical and cell biological approaches[17–20]. For example, the published cryo-EM structure of an autoinhibited, monomeric BRAF complex confirmed that a 14-3-3 dimer can bind simultaneously to the BRAF pS365 and pS729 sites and that the CRD makes critical contacts with both the RAF catalytic domain and the 14-3-3 dimer in the auto-inhibited state[17], consistent with previous studies implicating the CRD and 14-3-3 as key regulators of RAF autoinhibition[5–8]. Binding of the 14-3-3 dimer was also found to occlude both the membrane/ligand-binding region of the CRD and the dimer interface of the kinase domain[17], thus preventing spurious dimer formation and CRD–membrane contact. In contrast, the RBD was not sufficiently resolved in the autoinhibited complex to determine its exact position or orientation[17]. As a result, questions regarding whether the RBD interacts with other BRAF domains or the 14-3-3 dimer in the autoinhibited state remained open. Moreover, no insight was gained regarding how RAS binding relieves RAF autoinhibition and promotes the conformational changes required for dimerization and kinase activation. In addition, recent BRAF dimer structures have shown that the 14-3-3 dimer can bind simultaneously to two kinase domain protomers, but whether both protomers are catalytically active has been a matter of debate[17–20].

Using a mammalian cell expression system to isolate BRAF complexes, here we report the cryo-EM structures of two autoinhibited, monomeric BRAF complexes in which the RBD is well-defined. Only one of the autoinhibited complexes included MEK, but both contained a bound 14-3-3 dimer, and in both structures, the kinase domain was in a canonical inactive configuration and lacked ATP. In addition, we also report the cryo-EM structure of a BRAF dimer complex with both BRAF protomers bound to an ATP-competitive BRAF inhibitor and the kinase domains assuming the active conformation. The dimeric BRAF complex also included a dimer of 14-3-3 that bridged the C-terminal pS729 sites on each BRAF protomer, and the dimerized kinase domains exhibited an asymmetric orientation with respect to the 14-3-3 dimer. Notably, resolution of the RBD in our autoinhibited, monomeric BRAF structures provides key insights regarding how the orientation of the RBD allows access for RAS binding and how the interaction with RAS initiates the monomer-to-dimer transition required for RAF activation.

## Results

### Isolation and initial characterization of BRAF complexes.
To isolate BRAF complexes for cryo-EM analysis that would accurately reflect the regulatory mechanisms that take place in mammalian cells, we first established a human 293 FT cell line that stably expressed full-length, wild-type (WT) BRAF containing an N-terminal Halo tag (Fig. 1a). Using this cell line, monomeric BRAF complexes were isolated from quiescent, serum-depleted cells. Dimeric BRAF complexes were also isolated using cycling cells that were treated with the type I BRAF inhibitor SB590885 to promote RAF dimerization. BRAF complexes were collected from cell lysates using affinity chromatography and then separated by gel filtration chromatography to obtain homogenous samples containing recombinant BRAF in association with various endogenous proteins (Supplementary Fig. 1a–c). Analysis of the gel filtration fractions obtained from serum-depleted cells indicated a well-defined peak of monomeric BRAF that was comprised of two distinct physiological complexes, a larger complex containing 14-3-3 proteins and MEK (fraction 18) and a smaller complex containing 14-3-3 proteins but lacking MEK (fraction 19) (Supplementary Fig. 1a). To maximize purity and homogeneity of the MEK-bound complex, an additional protein preparation was generated from serum-depleted cells that had been treated with the MEK inhibitor CH5126766 to stabilize the BRAF–MEK interaction (Supplementary Fig. 1b). Analysis of the gel filtration fractions obtained from the RAF inhibitor SB590885-treated cells indicated a prominent peak near fraction 16 that represented a larger complex containing primarily BRAF and 14-3-3 proteins, with MEK and Hsp70 also observed (Supplementary Fig. 1c).

Mass spectrometry analysis of each of the three sample complexes revealed that 14-3-3ε and 14-3-3ζ constituted approximately 50% and 25%, respectively, of the 14-3-3 proteins present in both the monomeric and dimeric BRAF complexes (Supplementary Fig. 1d), a finding consistent with the expression level of these 14-3-3 isoforms in 293FT cells[21] and the propensity for 14-3-3ε to heterodimerize[22]. For the MEK-bound monomeric BRAF complex, MEK1 and MEK2 were present at a 54% and 46% ratio, respectively (Supplementary Fig. 1d). It should be noted that for the structures derived from cryo-EM, we used the human 14-3-3ζ isoform (PDB ID: 4FJ3) as a starting model for the 14-3-3 dimer and MEK1 (PBD ID: 3WGI) to represent MEK in the MEK-bound, autoinhibited complex.

### Structural analysis of the inhibitor-bound dimeric BRAF$_2$:14-3-3$_2$ complex.
A major advance in understanding RAF activation originates from studies showing that under most signaling conditions, dimerization of the kinase domains is required[23]. Many structures of isolated, dimeric BRAF kinase domains have been solved; however, recent structures of BRAF dimer complexes containing 14-3-3 proteins[17,18,20] have raised new questions regarding the mechanisms of dimer activation and whether both protomers in the dimer have catalytic activity. To further investigate these issues, we isolated BRAF dimer complexes from mammalian cells treated with the type 1 BRAF inhibitor SB590885 and obtained a cryo-EM structure of full-length, inhibitor-bound BRAF$_2$:14-3-3$_2$ at 3.9 Å resolution (Fig. 1b, c, and Supplementary Fig. 2a–c). It should be noted that despite the presence of MEK in the cryo-EM sample fraction, no density corresponding to MEK was observed in the 3D reconstructions. Additionally, the RBD and CRD as well as other regions of the N-terminal regulatory domain (residues 1-448) were also missing from the density map and are expected to be solvent-exposed and flexible. In our inhibitor-bound BRAF$_2$:14-3-3$_2$ structure, the BRAF kinase domains (KD) adopt a canonical back-to-back orientation with the αC-helices in the

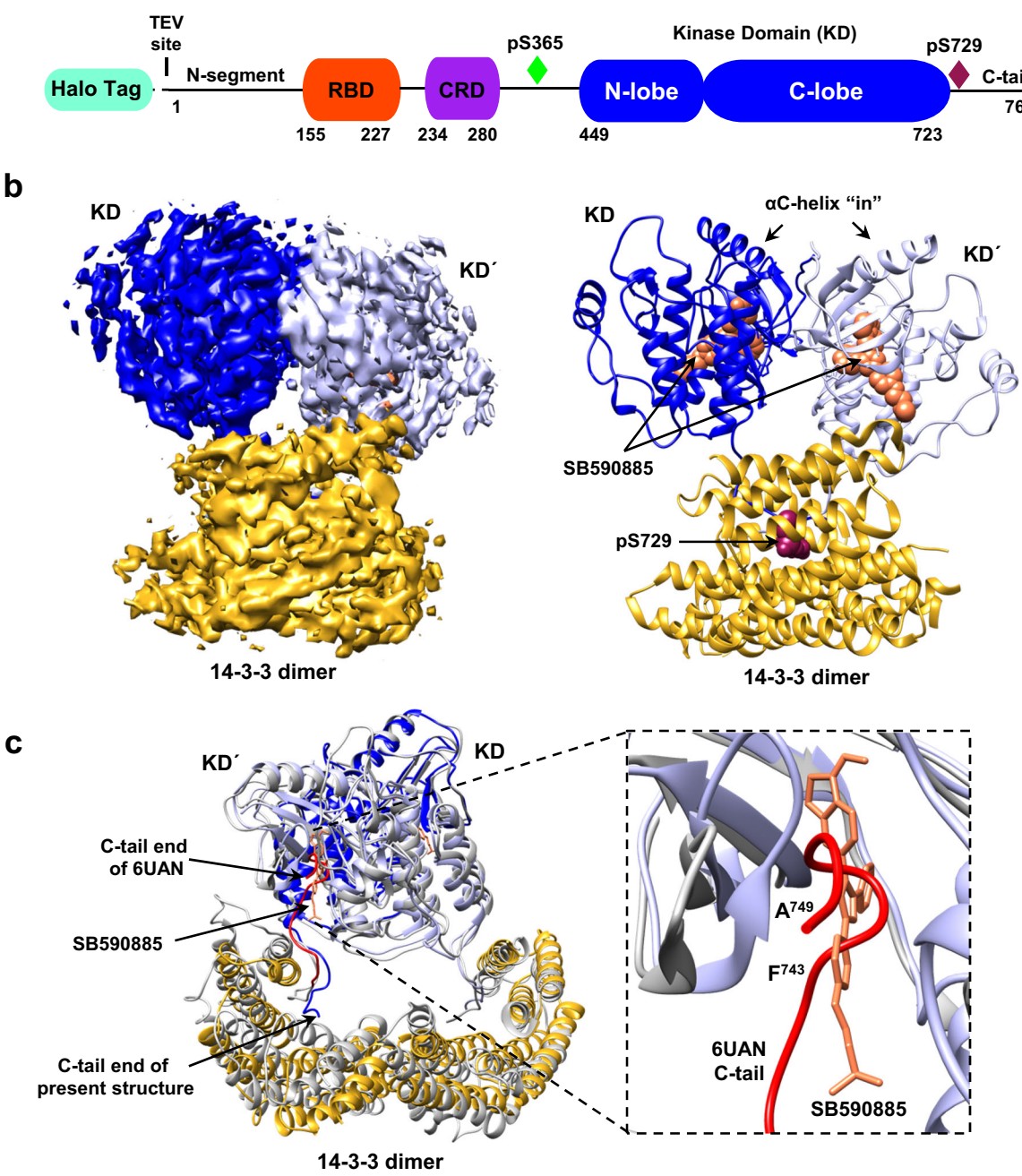

**Fig. 1 Cryo-EM structure of inhibitor-bound BRAF$_2$:14-3-3$_2$ dimer complex. a** BRAF domain organization is shown with color coding: RAS binding domain (RBD, red orange), cysteine-rich domain (CRD, purple), and the kinase domain (KD, blue). BRAF residues S365 (green) and S729 (dark red) serve as 14-3-3-binding sites when phosphorylated. **b** Cryo-EM density map at 3.9 Å resolution (left) and structure (right) of the BRAF$_2$:14-3-3$_2$ complex. BRAF domains are colored as in **a**, with one KD protomer in light blue and the other in dark blue. SB590885 is shown in coral and the 14-3-3 dimer is colored gold. **c** Superposition of the SB590885-liganded BRAF$_2$:14-3-3$_2$ structure (colored as in **b**) with the previously reported cryo-EM structure of the unliganded BRAF$_2$:14-3-3$_2$ complex in gray (PDB ID: 6UAN) is shown with a Cα RMSD of 1.54 Å for the overall structure. The inset demonstrates that residues F743-A749 in the C-tail end (in red) of the unliganded complex (PDB ID: 6UAN) and the BRAF inhibitor SB590885 (in coral) of the present structure occupy overlapping positions in the active site.

active "in" position and the regulatory (R) spines aligned[16,24] (Fig. 1b, c, and Supplementary Fig. 2d, e). As expected, a 14-3-3 dimer was bound to each C-terminal pS729 site, forming a stabilizing bridge between the KD protomers.

The orientation of the KD dimer with respect to the 14-3-3 dimer was asymmetric and similar to that previously described

for a cryo-EM structure of a ligand-free BRAF$_2$:14-3-3$_2$ complex[18] (Fig. 1c), with our overall structure superimposing well onto the ligand-free dimer structure (Cα root mean square deviation (RMSD) of 1.54 Å for the BRAF$_2$:14-3-3$_2$ regions). Notably, a major difference between the two structures is that for the unliganded BRAF$_2$:14-3-3$_2$ structure, the distal C-tail segment

(F743 to A749) of one KD protomer was found to insert into the active site of the other protomer, thus inhibiting its catalytic activity. Moreover, insertion of the C-tail segment into the active site was proposed to allow the "inhibited" protomer to act as a transactivator of the promoter from which the C-tail emanated and was also thought to account for the asymmetric orientation of the KD and 14-3-3 dimers.

In our inhibitor-bound $BRAF_2:14-3-3_2$ structure, interpretable density for both C-tails stops at residue 733, indicating that the C-tails (residues 734–766) are flexible after exiting the 14-3-3-binding pocket (Fig. 1b, c). In addition, density corresponding to the RAF inhibitor SB590885 was observed in both active sites (Fig. 1b, c, and Supplementary Fig. 2d) and overlapped with the position of the C-tail segment in the ligand-free dimer structure (Fig. 1c). This finding shows that insertion of the C-tail into the active site can be overcome by inhibitor binding and is consistent with recent biochemical analyses of active $BRAF_2:14-3-3_2$ complexes using differential scanning fluorimetry, microscale thermophoresis, and surface plasmon resonance approaches, which indicated that both ATP-binding sites are available for ligand binding even when the C-tail segment is present[19]. In addition, alignment of our inhibitor (SB590885)-bound $BRAF_2:14-3-3_2$ structure with the high-resolution crystal structure of inhibitor (GDC-0879)-bound BRAF KD (residues D432–R735)$_2$:14-3-3$_2$ complexes[20] yielded a Cα RMSD of 2.29 Å for the overall complex, with the main area of difference being one 14-3-3 protomer that is tilted closer to the BRAF kinase domain in the crystal structure, which may be due to the constraints of the crystallographic lattice (Supplementary Fig. 2f). For the GDC-0879-bound structure, inhibitor was also present in both active sites, and even though the KDs lacked the C-tail segments, an asymmetric orientation between the BRAF KD and the 14-3-3 dimer was observed. Taken together, our findings add support to the model that the asymmetry between the KD and 14-3-3 dimers is not determined by C-tail insertion and that the active sites of both protomers may bind ATP simultaneously to promote catalysis.

**Structural analysis of the monomeric BRAF:14-3-3$_2$:MEK and BRAF:14-3-3$_2$ complexes**. Over the years, less has been known regarding the structure of autoinhibited RAF monomers, with only one cryo-EM structure reported for full-length monomeric BRAF[17]. Here, using serum-depleted mammalian cells, we were able to isolate two monomeric BRAF complexes of sufficient homogeneity for structural analysis using cryo-EM, resulting in a BRAF:14-3-3$_2$:MEK structure of 3.7 Å resolution and a BRAF:14-3-3$_2$ structure of 4.1 Å resolution (Fig. 2a, b, and Supplementary Fig. 3a–c). Except for the presence or absence of MEK, the overall conformation of the monomeric BRAF structures aligned well, with a Cα RMSD of 0.95 Å for the corresponding BRAF:14-3-3$_2$ portions and a Cα RMSD of 0.99 Å for the BRAF KDs. In these structures, the BRAF monomer exhibits an autoinhibited conformation with a 14-3-3 dimer bound to the canonical pS365 and pS729 sites and the CRD making stabilizing contacts with the C-lobe of the kinase domain and with both protomers of the 14-3-3 dimer (Fig. 2a, b, and Supplementary Fig. 4). These results are consistent with the previously determined BRAF:14-3-3$_2$:MEK1$^{S218A,S222A}$ structure[17]; however, they show that BRAF can exist in the autoinhibited state in the absence of MEK. Moreover, our structures suggest that binding of the 14-3-3 dimer and BRAF intramolecular interactions are sufficient to maintain the auto-inhibited conformation, but that MEK binding may further stabilize the complex, as evidenced in the local resolution maps (Supplementary Fig. 3a, b).

The BRAF kinase domain in both monomer structures displayed the canonical kinase-inactive conformation[16,25], with the αC-helix in the "out" position and the R-spine broken (Fig. 2a and Supplementary Fig. 5a, and Supplementary Tables 1, 2). Residues in the activation segment (residues 598–602) formed an inhibitory turn to reinforce the extended outward shift of the αC-helix (Supplementary Fig. 5a). Although the ATP-binding pocket was unoccupied in our monomer structures, the structure of the KDs displayed a similar conformation as did recent inactive KD structures bound to ATP analogs, namely the ATP-γ-S-bound BRAF:14-3-3$_2$:MEK1$^{S218A,S222A}$ structure[17] (Cα RMSD of 0.90 Å for the KDs) and the BRAF KD (AMP-PCP):MEK1 structure[20] (Cα RMSD of 0.82 Å for the KDs) (Supplementary Fig. 5b, c, and Supplementary Tables 1, 2). Moreover, as was reported for the ATP-analog-bound structures, the N- and C-lobes of the KD in our ATP-free structures exhibited a closer orientation than is observed for the lobes of KDs bound to RAF inhibitors (Supplementary Fig. 5b, and Supplementary Table 2)[17,20,26,27]. These findings indicate that the ATP-binding pocket of the BRAF KD is stable in the "apo" nucleotide-free state and that while ATP-binding may be required to form the compact configuration of the N- and C-lobes as well as the autoinhibited BRAF conformation, these states can exist in the absence of bound ATP (Supplementary Fig. 5b, c, and Supplementary Tables 1, 2).

For the larger monomeric BRAF complex containing MEK, the active sites of MEK and the BRAF KD are in a canonical face-to-face orientation, with the C-lobes of both kinases making extensive contacts and the activation segments running in an antiparallel manner (Fig. 2a and Supplementary Fig. 5d). As expected, MEK was in the inactive conformation with its αC-helix in the "out" position and density for the MEK inhibitor CH5126766 visible within the allosteric-binding site (Supplementary Fig. 5d). In addition, residues of the αA-helix of MEK, which acts in a negative regulatory manner to stabilize the inactive conformation[28], were resolved and lay against the base of the MEK αC-helix (Supplementary Fig. 5d).

Consistent with the previously published autoinhibited BRAF:14-3-3$_2$:MEK1$^{S218A,S222A}$ structure, the N-terminal segment of BRAF (amino acids 1–155) and the linker sequences between the CRD and the pS365 site (amino acids 281–359), and between the pS365 site and the kinase domain (amino acids 371–448) are not resolved in our monomeric structures, likely reflecting the flexibility of these regions. In contrast, the RBD is well-defined in our autoinhibited BRAF:14-3-3$_2$:MEK and BRAF:14-3-3$_2$ structures (Fig. 2a, b and Supplementary Fig. 4), revealing the position and orientation of this critical domain.

**RBD orientation and contacts in the autoinhibited monomeric BRAF complexes**. A distinct feature of the BRAF structures reported here is the resolution of the RBD in the context of the full-length, autoinhibited BRAF monomer. RBDs are known to have a conserved ubiquitin-like structure[29] containing five beta-strands and two–three alpha helices. As shown in Fig. 2a, the BRAF RBD in both monomeric structures sits on top of the 14-3-3 protomer bound to the C-terminal pS729 site and adjacent to the C-lobe of the KD, with N163 of the RBD being within hydrogen bond distance to S679 of the KD (Supplementary Fig. 6a). No RBD contact is observed with the CRD or with MEK in the MEK-bound structure. Notably, an extensive contact surface of ~435 Å$^2$ is observed between the RBD and the 14-3-3 protomer, with the electrostatic charge distributions of the interface surfaces having considerable complementarity (Fig. 3a). The α1-helix of the RBD is oriented along the interface and interacts with the α8-helix, α9-helix, and loop 8 of 14-3-3 (Fig. 3b

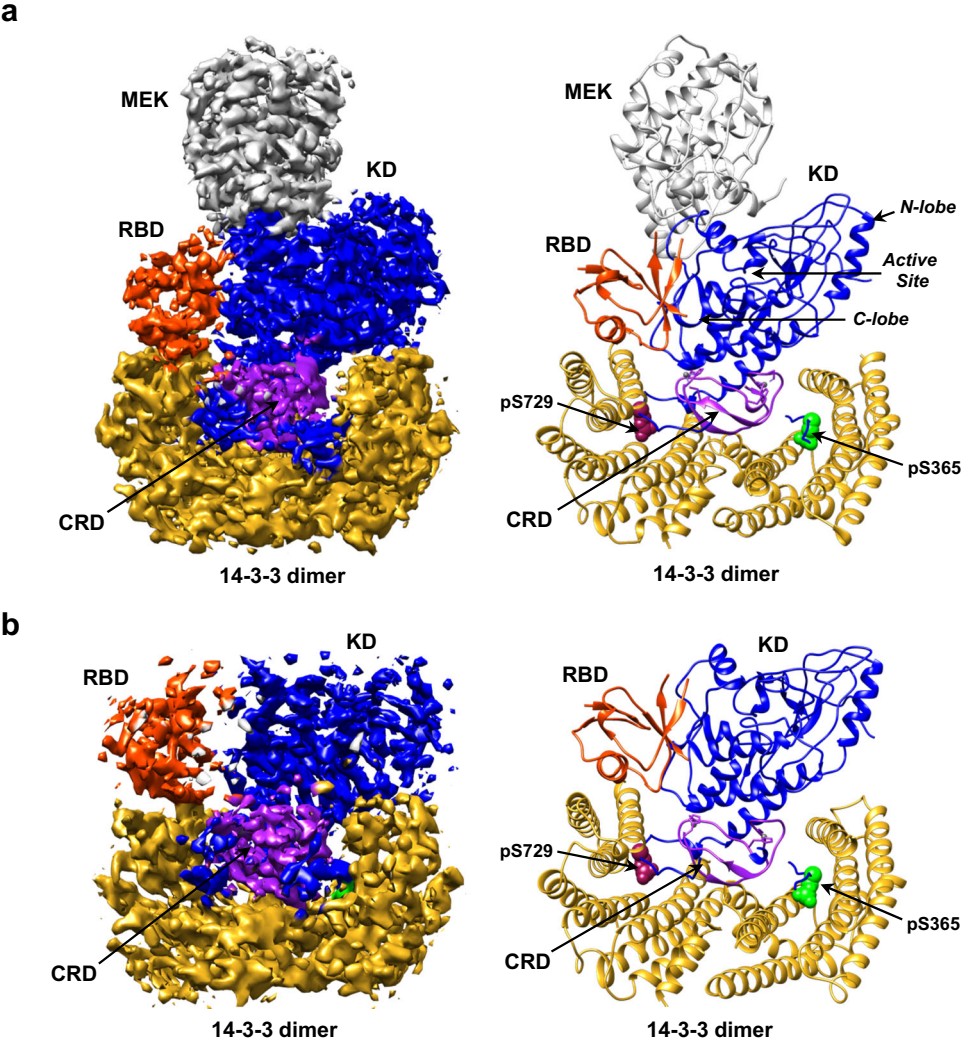

**Fig. 2 Structures of the autoinhibited, monomeric BRAF:14-3-3₂:MEK and BRAF:14-3-3₂ complexes. a** Cryo-EM density map at a 3.7 Å resolution (left) and structure of the BRAF:14-3-3₂:MEK (right) autoinhibited complex. **b** Cryo-EM density map at a 4.1 Å resolution (left) and structure of the BRAF:14-3-3₂ (right) autoinhibited complex. Regions of the BRAF monomer and the 14-3-3 dimer are colored as previously described and MEK is shown in gray.

and Supplementary Fig. 6b). The 14-3-3 residues involved in this interface are conserved in all human 14-3-3 family members, indicating that a similar interface would be predicted regardless of the isoform composition of the 14-3-3 dimer. In contrast, two RBD residues at the center of the RBD:14-3-3 interface, M186 and M187, are not conserved, being replaced by a lysine and valine in CRAF and ARAF (Fig. 3b).

Because the M186 and M187 residues have the potential to make numerous contacts with 14-3-3 at the interface and given that contacts between 14-3-3 and the CRD are known to play a key role in maintaining the autoinhibited state, we next took a mutational approach to determine whether these RBD residues also contribute to RAF autoinhibition. M186 and M187 were mutated either to lysine and valine (as in CRAF and ARAF), to smaller alanine residues, to bulkier but still hydrophobic tryptophanes, or to negatively charged glutamic acid residues, following which the effect of these mutations on RAF autoinhibition was assessed using a proximity-based NanoBRET assay (Fig. 3c). In this assay, BRAF is split into two segments, with the regulatory domain (1-435) tagged with an acceptor fluorophore (BRAF^REG-Halo) and the kinase domain (436-766) tagged with an energy donor (NanoLuc-BRAF^KD). When the two segments interact to form an autoinhibited complex that is stabilized by 14-3-3 dimer binding,

the donor and acceptor are brought within range for energy transfer to occur (<100 Å), resulting in the generation of a BRET signal. As a control, we found that when the CRD of BRAF^REG contained the RASopathy-associated T241P mutation, which reduces the autoinhibitory effect of the CRD, the BRET signal was decreased ~50% (Fig. 3c). Analysis of the RBD mutants revealed that the M186K/M187V mutant generated a BRET signal that was equivalent to WT-BRAF^REG. In contrast, a reproducible 10% decrease in signal was observed for the M186W/M187W mutant, whereas a 17% increase in signal was observed for the M186A/M187A mutant. Surprisingly, the glutamic acid substitutions did not disrupt BRAF autoinhibitory interactions but rather result in a reproducible 5–10% increase in the BRET signal, which may reflect potential interactions with R222 in the 14-3-3 α9-helix. Both M186A/M187A-BRAF^REG and M186E/M187E-BRAF^REG were also found to be more effective at suppressing MEK activation mediated by the isolated BRAF^KD protein than was WT-BRAF^REG (Fig. 3d), indicating an enhanced ability to form a stable, autoinhibited BRAF complex. In addition, when incorporated into full-length BRAF, the M186W/M187W mutant exhibited increased biological activity in focus forming assays (Fig. 3e). Moreover, the activity of the RBD M186W/M187W and CRD T241P mutants in the focus forming assay correlated with the

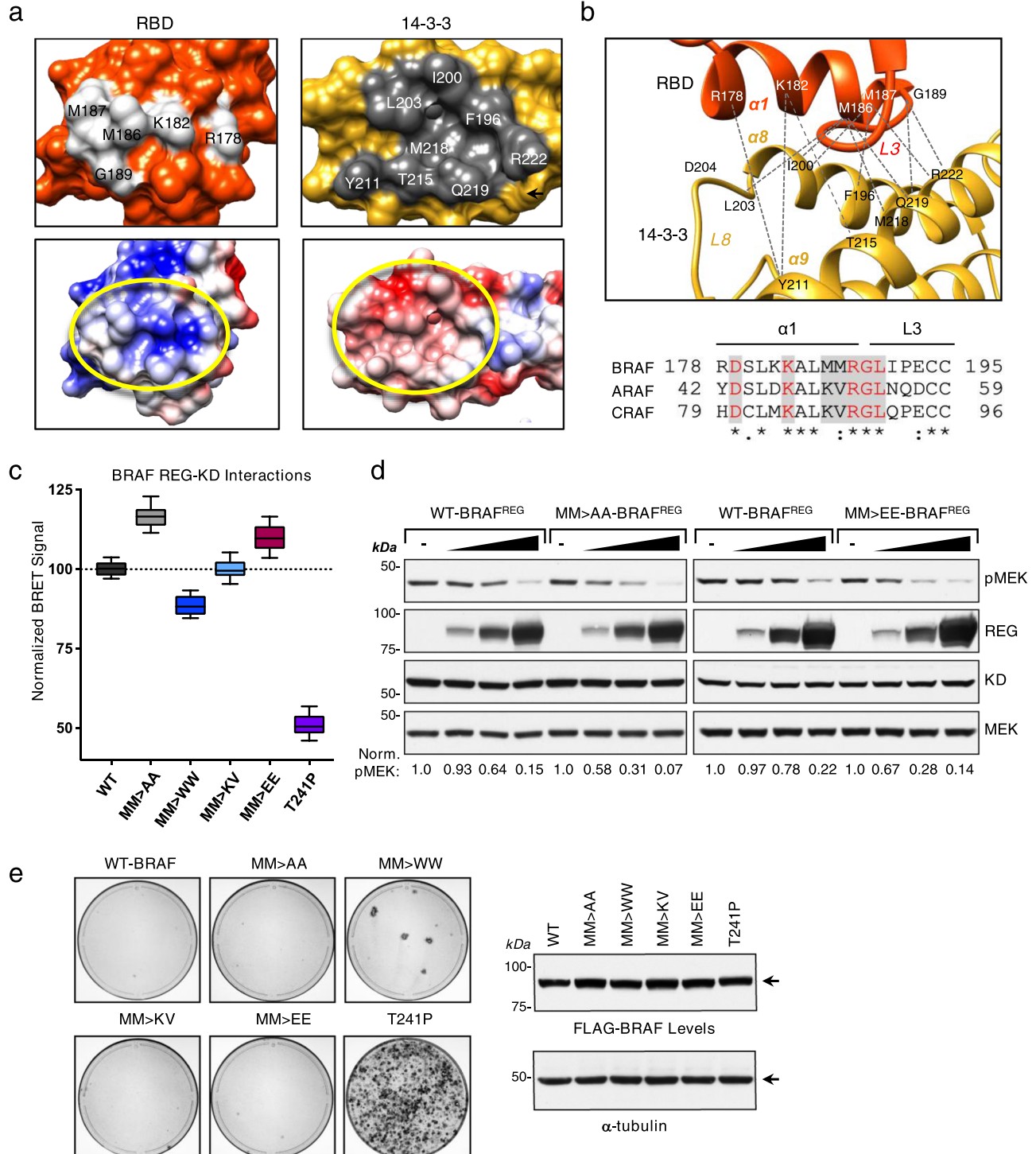

ability of these mutations to disrupt RAF autoinhibition in the BRET assay. Thus, while the CRD plays the predominant role in autoinhibition, these findings indicate that M186/M187 and the RBD also contribute to the maintenance of the autoinhibited state.

**Effects of RAS binding on the BRAF RBD:14-3-3 interface.** Under normal signaling conditions, the RAF activation process begins when an autoinhibited RAF monomer interacts with GTP-bound RAS[9–11]. Previously determined crystal structures of RAS–RBD complexes have shown that the β2-strand and the α1-helix of the RBD interact with the β2-strand and the switch I region of RAS[30,31] and that RAS–RBD binding involves ionic and

hydrogen bonds as well as other Van der Walls interactions[32]. In our autoinhibited BRAF complexes, the RAS-binding surface of the RBD faces away from the CRD and is oriented towards a space that we term the "RAS pocket" (Fig. 4a). This pocket is flanked by 14-3-3, the C-lobe of the KD, and MEK in the BRAF:14-3-3₂:MEK structure (Fig. 4a, b). Critical basic residues of the RBD, R158, R166, K183, and R188, are exposed and face towards the RAS pocket, ready to form ionic bonds with acidic residues E31, D33, E37, and D38 in RAS switch I (Fig. 4b).

To demonstrate that the RBD in our autoinhibited BRAF monomeric complexes is accessible for RAS binding, fluorescence polarization assays were conducted in which the purified

**Fig. 3 Analysis of the RBD:14-3-3 interface. a** Region of RBD:14-3-3 contact, with RBD and 14-3-3 residues at the contact interface depicted as light gray and dark gray, respectively. Interacting residues are labeled (top). Electrostatic surface representation of the RBD and 14-3-3, with blue and red representing positively and negatively charged areas, respectively. Regions of contact between the RBD and 14-3-3 lay within the yellow circles (bottom). **b** The 14-3-3 α8-helix, α9-helix, and the connecting loop 8 contact the α1-helix and loop 3 of the BRAF RBD. The RBD is colored in orange, 14-3-3 in gold and the interacting residues are labeled (top). Sequence alignment of RBD residues in the α1-helix and loop 3 of human BRAF, CRAF, and ARAF (bottom). Residues at the RBD:14-3-3 interface are denoted by the shaded gray box, and identically conserved residues are shown in red. Symbols under the alignment denote the degree of conservation as follows: (*) indicates positions that have a fully conserved residue, (:) indicates conservation between groups with strongly similar properties, and (.) indicates conservation between groups with weakly similar properties. **c** NanoBRET assay monitoring the interaction between WT or mutant BRAF[REG]-Halo proteins and Nano-BRAF[KD] in live cells to determine the effect of the indicated mutations on the ability of BRAF to form a stable, autoinhibited complex. BRET signals (normalized to WT set at 100) of quadruplicate wells from five independent experiments were used to generate a box and whiskers plot, with the center line representing the median, the box limits extending from the 25th to 75th percentile, and the whiskers extending from the minimum to maximum values. **d** Lysates of 293FT cells transiently expressing Nano-WT-BRAF[KD] alone or co-expressing Nano-WT-BRAF[KD] with increasing amounts of WT-, M186A/M187A-, or M186E/M187E-BRAF[REG]-Halo were examined by immunoblot analysis for WT-BRAF[REG]-Halo, Nano-WT-BRAF[REG], and pMEK levels. Blots are representative of three independent experiments with similar results. **e** NIH-3T3 cells were infected with retroviruses encoding the indicated WT or mutant FLAG-BRAF[FL] proteins. Three weeks post-infection, foci were visualized by methylene blue staining. Stained plates are representative of three independent experiments with similar results. Source data are provided as a Source Data file.

---

BRAF:14-3-3$_2$:MEK and BRAF:14-3-3$_2$ complexes were titrated onto GFP-KRAS that had been pre-loaded with a GTP analog. As shown in Fig. 4c, both complexes exhibited a high degree of binding and had affinities in the nanomolar range. RAS–RAF binding could also be demonstrated in KRAS[G12V] pull-down assays using the purified BRAF:14-3-3$_2$:MEK and BRAF:14-3-3$_2$ complexes (Fig. 4d). As expected, mutation of any of the RBD basic residues that form ionic bonds with KRAS (R158A, R166A, K183, and R188L) significantly disrupted KRAS[G12V] binding in the pull-down assays (Fig. 4e), confirming that RBD contact is essential for the RAS–BRAF interaction.

Although it has been known that binding to RAS relieves RAF autoinhibition and allows RAF to dimerize, the precise molecular details for how this is achieved have been unclear. Based on the known KRAS:RBD interface[32], we superimposed KRAS onto our autoinhibited BRAF structures such that all of the ionic interactions needed for high-affinity binding could form. Strikingly, in this position, a steric clash would occur between residues in the KRAS α1-helix and switch I region and the α8- and α9-helices of 14-3-3 that lie beneath the RBD and form the RBD:14-3-3 interface (Fig. 4f). Specifically, RAS residues I21, Q22 (α1-helix), Q25-E31 (SI region), K42 and V45 (β2 sheet) would clash with 14-3-3 α8-helix residues C189, A192-E198, I200, A201, and α9-helix residues R222, L225, T226 and S230 (Supplementary Fig. 6c). Moreover, this region of 14-3-3 has a notable negative charge and, while complementary with the positively charged RBD interface, would cause electrostatic repulsion with negatively charged residues in KRAS switch I (Fig. 4f). In particular, KRAS switch I residues D30 and E31 would be brought in close proximity to D197 and E198 of 14-3-3. Thus, we propose the following model in which RAS-BRAF binding is a dynamic process and begins with the recruitment of the autoinhibited BRAF monomer to the membrane through interactions between RAS and the exposed basic residues of the RBD. Formation of the ionic bonds with RAS would generate steric clashes and electrostatic repulsion at the RBD:14-3-3 interface that would initiate a rearrangement in 14-3-3 dimer binding, resulting in the exposure of additional RBD residues involved in full RAS–RBD contact. We predict that the full spectrum of RAS–RBD interactions would dislodge the RBD from the autoinhibited complex as well as the CRD, due to the short linker between these two domains, thereby enabling the CRD to rotate and make contact with RAS and the plasma membrane. This structural reoganization, in turn, would promote the release of 14-3-3 from the pS365 site and expose the BRAF dimer interface for dimer formation and catalytic activation.

In support of this model, our results below indicate that the RBD M186/M187 residues at the center of RBD:14-3-3 interface

serve a dual role by mediating contact with 14-3-3 in the autoinhibited state and by contributing to RAS-binding interactions upon RAS activation. These methionines are lysine and valine in CRAF, and structural studies indicate that K87 and V88 of CRAF contribute to full RAS:RBD contact by forming extensive non-electrostatic interactions with KRAS residues I24, Q25, and Y40[32]. Although M186 and M187 in BRAF are partially or fully occluded by 14-3-3 in the autoinhibited state, the steric clash caused by initial RAS:RBD contact would directly impact 14-3-3 residues that interact with these methionines, perhaps resulting in their exposure for contact with RAS. Therefore, to address whether these BRAF residues contribute to the RAS–RBD interaction, the M186/M187 mutants were assessed for binding to KRAS[G12V] in co-immunoprecipitation experiments and in a BRET assay that monitors RAS-RAF binding in live cells. In particular, the BRET assay allows for RAS-RAF binding levels (BRET$_{max}$) and for the affinity of the interaction (BRET$_{50}$) to be quantified when saturation curves are performed[33]. From these assays, we found that mutation of M186/M187 to alanine or glutamic acid residues reduced binding to KRAS[G12V] (Fig. 5a, b), with the glutamic acid substitutions having the most pronounced effect. In contrast, when M186/M187 were mutated to bulkier tryptophan residues (M186W/M187W) or changed to lysine and valine (M186K/M187V), as in CRAF, increased levels of binding to KRAS[G12V] were observed in both the BRET and co-immunoprecipitation assays (Fig. 5a, b).

To test whether the modulation in binding reflects direct RAS–RBD contact, RBD pull-down experiments were performed using isolated WT or mutant BRAF RBD domains. As was observed for the full-length BRAF proteins, the M186W/M187W and the M186K/M187V mutants exhibited increased binding to KRAS[G12V] (Fig. 5c), whereas lower binding levels were observed for the M186A/M187A mutant, and little to no binding was detected for the M186E/M187E mutant, indicating that the mutations were directly influencing interactions between the BRAF RBD and KRAS[G12V]. Modeling of the amino acid substitutions (Fig. 5d, e) into the K87/V88 positions of a CRAF RBD:RAS structure[32] further supports these findings in that while the KV/MM substitutions would change some of the backbone and side-chain interactions, non-electrostatic interactions would still be predicted. With the KV/WW mutations, the bulkier tryptophan residues would have the potential to form additional interactions, whereas the alanine substitutions would result in the loss of all side-chain contacts. For the KV/EE mutations, the negatively charged glutamic acid residues would cause electrostatic repulsion with RAS residues E31 and D33 and be inhibitory to binding. Thus, taken together our mutational analysis indicates

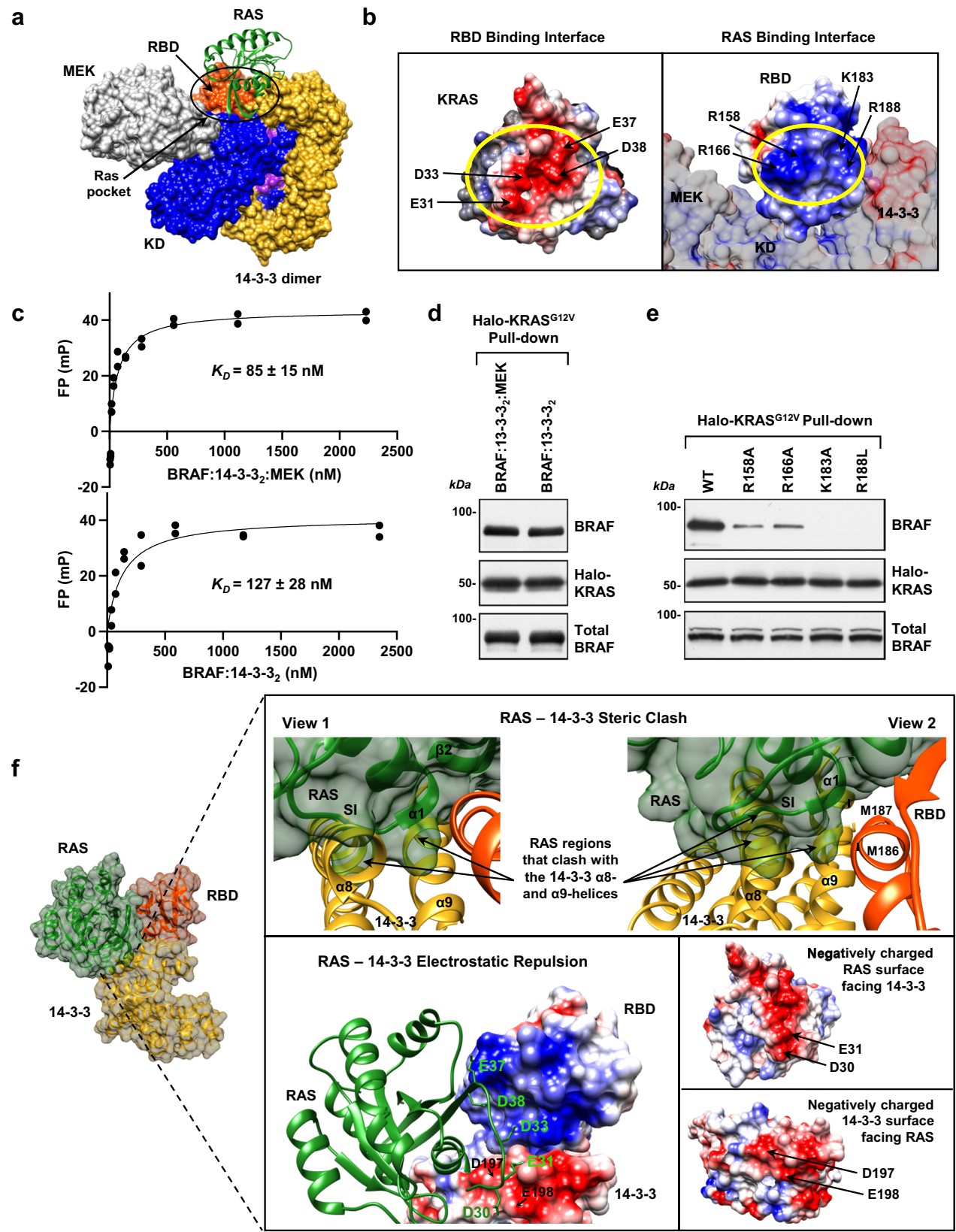

that these central residues in the BRAF RBD:14-3-3 interface serve dual functions in first contributing to the autoinhibited state through interactions with 14-3-3 and then in participating to the full spectrum of RAS–RBD interactions that facilitate the conformation changes needed to disrupt the autoinhibited state.

## Discussion

It is well established that under most signaling conditions binding to activated RAS is required for RAF dimerization and activation; however, the structural details for how RAS binding allows autoinhibited RAF monomers to assume an active dimer

**Fig. 4 RAS, BRAF RBD, and 14-3-3 interactions. a** Superimposition of KRAS (from KRAS:CRAF_RBD structure PDB ID: 6XI7, colored in green) onto the autoinhibited BRAF:14-3-3$_2$:MEK complex is shown to demonstrate the fit of KRAS into the "RAS pocket" for RAS:RBD binding. **b** Electrostatic surface representation of the RAS:RBD-binding interface (yellow circle), with critical ionic bond-forming residues indicated on the surfaces of KRAS (PDB ID: 6XI7, left) and the RBD of the BRAF:14-3-3$_2$:MEK complex (right). **c** Binding affinities of the BRAF:14-3-3$_2$:MEK (top) and BRAF:14-3-3$_2$ (bottom) complexes to KRAS(GppNHp) were determined in fluorescence polarization assays. Data points present duplicate wells. The reported $K_D$ is the mean of three independent experiments. **d** Purified BRAF:14-3-3$_2$:MEK and BRAF:14-3-3$_2$ complexes were evaluated for their ability to bind Halo-tagged KRAS$^{G12V}$ in pull-down assays. Blots are representative of two independent experiments with similar results. **e** WT or BRAF full-length proteins containing mutations in key basic RBD residues mediating ionic bond interactions with RAS were examined for binding to Halo-KRAS$^{G12V}$ in pull-down assays. Blots are representative of three independent experiments with similar results. **f** Superimposition of KRAS onto the autoinhibited BRAF:14-3-3$_2$:MEK complex indicates potential steric clash and electrostatic repulsion between RAS and the 14-3-3 protomer at the RBD:14-3-3 interface upon full RAS:RBD contact. The steric clash (top insets) would occur between RAS α1-helix, switch I (SI) and β2 sheet (ribbon and surface in green) and the α8- and α9- helices of 14-3-3 (ribbon in gold). In the same region, similarly charged residues in KRAS, D30, and E31, and 14-3-3, D197, and E198, would cause electrostatic repulsion (bottom inset, RAS ribbon structure in green superimposed onto the electrostatic surface representation of the RBD and 14-3-3). Source data are provided as a Source Data file.

conformation has been unclear. In this study, we successfully obtained cryo-EM structures of autoinhibited, monomeric BRAF complexes (BRAF:14-3-3$_2$:MEK and BRAF:14-3-3$_2$ at 3.7 and 4.1 Å resolution, respectively) and an RAF inhibitor-bound dimeric complex (BRAF$_2$:14-3-3$_2$ at 3.9 Å resolution). These complexes were isolated from a mammalian cell expression system such that components of the complex would be subject to normal cellular regulation and more closely represent physiological signaling complexes. The autoinhibited BRAF structures show that prior to signaling events, BRAF and its substrate MEK can exist as a preassembled BRAF:14-3-3$_2$:MEK complex; however, the interaction with MEK is not required for BRAF to maintain the autoinhibited conformation. This finding is consistent with a previous study which found that BRAF was still present in the flow-through of human 293 cell lysates where MEK had been immunodepleted, suggesting a dynamic equilibrium between MEK-bound and MEK-free BRAF complexes[34]. Notably, both of our autoinhibited BRAF structures had clearly resolved RBDs, revealing the position and orientation of this critical domain and providing valuable insight regarding how RAS binding facilitates the monomer to dimer transition.

The position and orientation of the RBD in the context of the autoinhibited BRAF complexes indicates that the RBD is accessible for RAS binding, with key basic residues involved in RAS contact exposed. However, it should be noted that the large BRAF specific, N-terminal segment that precedes the BRAF RBD was not resolved in our structures. Thus, it is possible that this region may represent an additional level of regulation in terms of the RAS–BRAF interaction, as has been suggested in recent live-cell studies analyzing the binding preferences of the RAS and RAF family members[33]. Nonetheless, both the purified BRAF:14-3-3$_2$:MEK and BRAF:14-3-3$_2$ complexes were fully competent to bind GTP-bound KRAS in fluorescence polarization assays and displayed high-affinity binding. Our autoinhibited structures also revealed a significant contact interface between the RBD and the 14-3-3 protomer bound to the C-terminal pS729 site. The interface surfaces had charge complementarity, with the RBD surface containing several positively charged residues and the 14-3-3 surface having a predominantly negative charge. The 14-3-3 residues that comprise the interaction surface are conserved in all 14-3-3 family members, indicating that the surface would be the same irrespective of which isoform was bound.

Strikingly, our results indicate that two RBD residues at the center of RBD:14-3-3 interface, M186/M187, also contribute to RAS binding and may play a key role in facilitating the conformational changes in 14-3-3 binding that are needed for dimer formation. Recent crystal structures of CRAF RBD and CRAF RBD-CRD in complex with KRAS indicate that CRAF residues (CRAF K87/V88) at analogous positions to the BRAF

methionines make van der Waals contacts with residues I24/Q25/Y40 in RAS[32]. Although no structure of RAS in complex with the BRAF RBD currently exists, our mutational analysis of the BRAF RBD M186/M187 residues suggests that alterations in these residues can modulate KRAS binding. Thus, integrating our autoinhibited BRAF structures into what is known regarding the RAS–RBD interaction strongly supports a model whereby RAS binding is a highly dynamic process (Fig. 6). The initial contact with activated RAS is likely made by exposed basic residues in the BRAF RBD (R158, R166, K183, R188). As RAS is engaged to form the high-affinity ionic bonds, a steric clash and electrostatic repulsion between RAS and 14-3-3 would occur at the RBD:14-3-3 interface, leading to the exposure of additional RBD residues involved in the full spectrum of RAS–RBD interactions. We predict that together these events will dislodge the RBD and in turn the CRD from the autoinhibited structure, thus putting further strain on the bound 14-3-3 dimer that ultimately results in its release from the pS365 site. The freed CRD could then contact RAS and the plasma membrane to further stabilize RAS–RAF complexes and aid in orienting the KD for dimer formation. It should be noted that crystal structures of RAS bound to a truncated CRAF protein containing only the RBD-CRD have recently been reported (HRAS:CRAF_RBD-CRD[35] and KRAS:CRAF_RBD-CRD complexes[32]) and demonstrate significant contact of the CRAF CRD with RAS. Based on the position and orientation of the RBD and CRD in our autoinhibited BRAF structures, it appears that if the BRAF CRD makes similar contacts with activated RAS, the CRD would need to rotate as it is dislodged from the autoinhibited complex in order to assume the final RAS-binding state. More specifically, when the bound RBD-CRD structure is overlayed onto our autoinhibited structure, with the alignment based on the CRD position, the RBD of the CRAF_RBD-CRD structure would lie outside the cryo-EM density map (Supplementary Fig. 4g). Thus, the RAS:CRAF_RBD-CRD structure may be more representative of the final RAS-BRAF binding conformation rather than the initial contact between RAS and the autoinhibited BRAF complexes.

Not surprisingly, given the critical role of the membrane environment in forming the active dimeric RAF complex, the entire regulatory region of BRAF was unstructured in our dimeric BRAF$_2$:14-3-3$_2$ complex. As RAF inhibitor treatment was used to facilitate the isolation of dimer complexes, SB590885 was detected in the ATP-binding pocket of both KD protomers. Each KD protomer also interacted with a protomer of a 14-3-3 dimer, which was bound to the C-terminal pS729 site. The orientation between the KD and 14-3-3 dimers was asymmetric and similar to that observed in the other structures of dimerized BRAF KDs bound to 14-3-3[18–20]. This asymmetry was initially proposed to

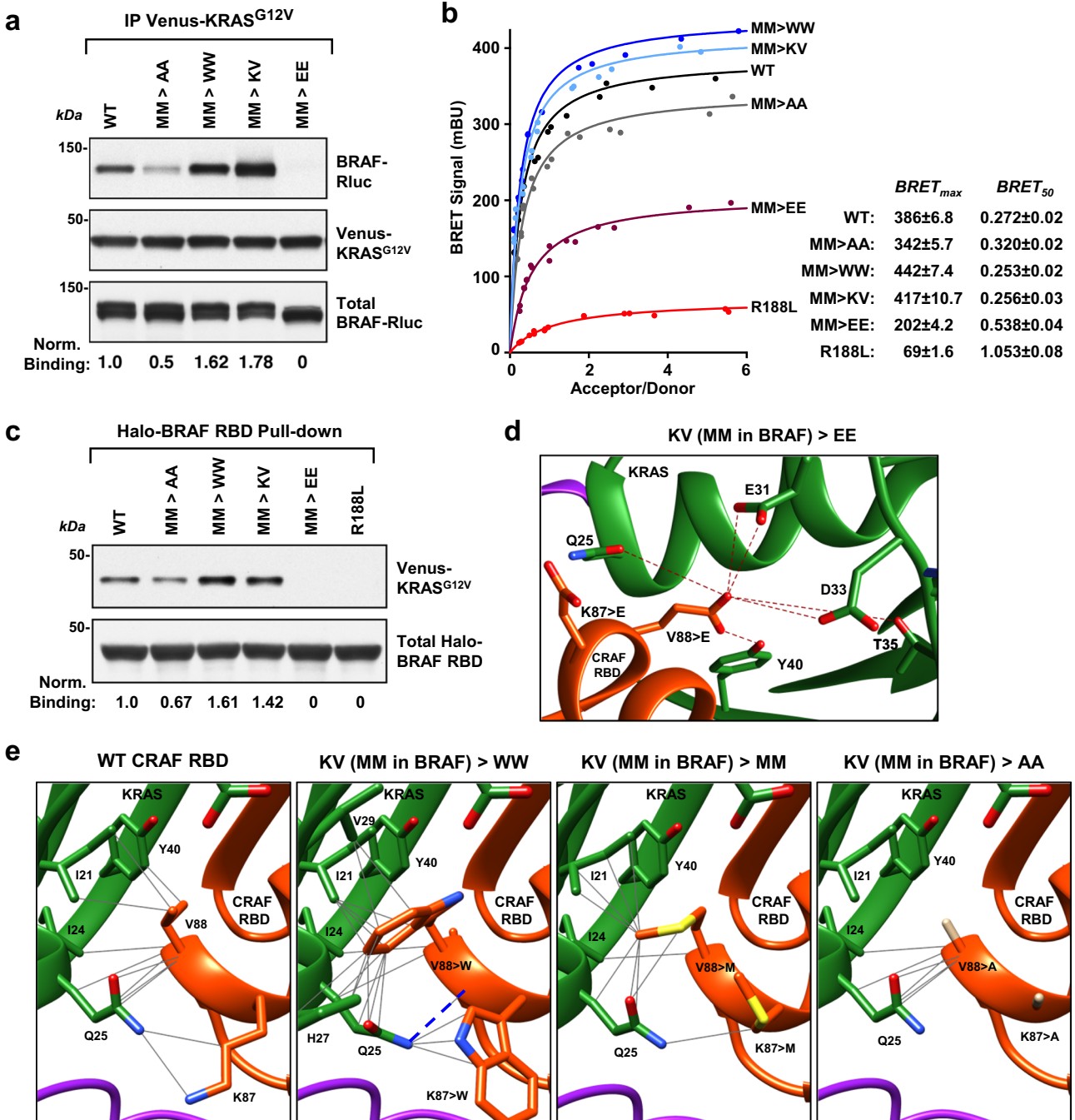

**Fig. 5 RBD residues at the 14-3-3 interface contribute to RAS binding. a** WT-BRAF-Rluc and the indicated M186/M187 mutants were assessed for binding to Venus-KRAS$^{G12V}$ in co-immunoprecipitation assays. Lysates were also monitored for BRAF-Rluc protein levels. Blots are representative of three independent experiments with similar results. **b** BRET saturation curves are shown examining the interaction of WT- or M186/ M187 mutant BRAF-Rluc proteins with Venus-KRAS$^{G12V}$ in live cells. As a control, an RBD protein containing the R188L mutation that disrupts the RAF–RAS interaction was also evaluated. BRET$_{max}$ and BRET$_{50}$ values are listed ±standard error on the right. Saturation curves were repeated three times with similar results. **c** Binding of Venus-KRAS$^{G12V}$ to WT or mutant Halo-BRAF RBD proteins was assessed in pull-down assays. Blots are representative of three independent experiments with similar results. **d** Modeling the effect of substituting the CRAF K87/V88 amino acids for K87E/V88E, which would generate electrostatic repulsion (indicated by dark red lines) with polar (Q25, T35, and Y40) and negatively charged (E31 and D33) KRAS residues. **e** Modeling the effect of substituting the CRAF K87/V88 amino acids for K87W/V88W, K87M/V88M, or K87A/V88A in the RBD:RAS structure (KRAS:CRAF_RBD_CRD structure PDB ID: 6XI7, colored in green for KRAS, orange for CRAF RBD, and violet for CRAF CRD). Gray lines indicate potential non-electrostatic interactions and blue dashed lines shows potential hydrogen bonds. Source data are provided as a Source Data file.

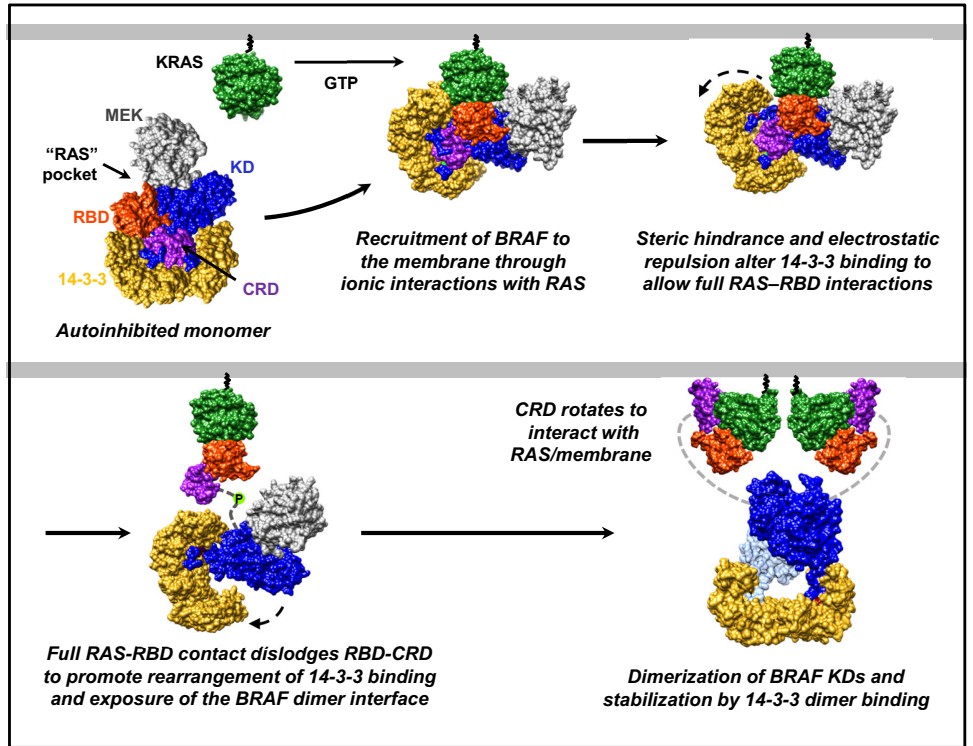

**Fig. 6 Model for BRAF monomer to dimer transition.** Upon RAS activation, the autoinhibited BRAF monomer is recruited to the plasma membrane through direct ionic interactions between the BRAF RBD and the switch I region of RAS. As RAS forms the full spectrum of RBD contacts, steric clashes, and electrostatic repulsion occur between RAS and 14-3-3, resulting in conformational changes that dislodge the RBD/CRD and promote a rearrangement in 14-3-3 dimer binding, thus exposing the pS365 site and the BRAF dimer interface. The dislodged CRD rotates to interact with the membrane and RAS to further stabilize the interaction. The exposed KD can then dimerize and assume the active catalytic conformation that is stabilized by 14-3-3 dimer binding to the pS729 sites on each BRAF protomer.

result from the insertion of a C-tail segment from one BRAF KD protomer into the active site of the other[18]; however, the fact that a similar orientation has been observed when the active sites are fully occupied by RAF inhibitors or when the KDs lack the C-terminal segments has argued against this model[19,20].

Finally, a comparison of the KD conformation in inhibitor-bound BRAF dimer complexes with that of our autoinhibited, monomeric BRAF structures is in agreement with previous observations that the N- and C-lobes of the inhibitor-bound dimerized complexes are in a more open configuration[17,20]. Interestingly, in our monomeric BRAF structures, this compact kinase domain configuration, which is known to put the dimer interface in an unfavorable position to form the N- to C-lobe antiparallel interactions needed to form dimers, was observed in the absence of ATP binding, differing from the previously reported structures exhibiting this conformation[17,20]. Nonetheless, our findings are in agreement with the model that identifying compounds or strategies to stabilize this compact configuration may have therapeutic potential in blocking aberrant RAF dimerization.

In summary, our studies contribute to the understanding of the BRAF signaling process by providing biochemical and structural information regarding different BRAF states. Further investigation is still needed to determine the mechanistic details for the transition between states and how these transitions are altered by events such as phosphorylation, dephosphorylation, changes in the signaling environment, and interactions with other binding partners. Moreover, it will be important to determine if other RAF family members have similar monomer and dimer architectures and whether heterodimer complexes may differ. This knowledge will provide a more mechanistic understanding of the

signaling process and may aid in the development of more effective treatment strategies for RAS- and/or RAF-driven disease states.

## Methods

**Reagents and resources.** Resources used in this study are listed in Supplementary Table 3. Plasmids and cell lines are available for use upon request to the corresponding authors.

**Cell lines and culture conditions.** 293FT, 293T, HeLa, NIH-3T3, and Phoenix-Eco cells were cultured in DMEM supplemented with 10% fetal bovine serum (FBS), 2 mM L-glutamine, and 1% penicillin/streptomycin. All cells were cultured at 37 °C with 5% $CO_2$.

**Generation of recombinant lentivirus and stable 293FT cells expressing Halo-BRAF^WT.** Recombinant lentivirus particles were generated by co-transfecting psPAX2 and pMD2 with the pLenti Halo-BRAF^WT construct into 293T cells using the X-tremeGENE9 transfection reagent. 72 h post-transfection, the virus-containing supernatant was collected, centrifuged twice at 935 × $g$ for 10 min to remove any cellular debris, and then stored at −80 °C. 293FT cells were infected with viral supernatants containing 8 μg/mL polybrene for 24 h, following which growth media containing 2 μg/mL puromycin (selection media) was added. The selection media was changed every 3 days until resistant cells were obtained.

**Affinity purification of BRAF complexes and size exclusion chromatography.** Halo-BRAF^WT 293FT cells were plated into 45, 10 cm tissue culture dishes at a concentration of $2 \times 10^6$ cells/dish and allowed to grow to confluence (~3 days after plating). For BRAF monomer samples, media was left unchanged until collection to allow for serum depletion. For BRAF dimer samples, media was replaced the day prior to collection with fresh growth media containing 2 μM SB590885. On the day of collection, media was removed by suction from each plate and the cells were washed twice with 5 mL cold PBS. Cells were then lysed in Triton lysis buffer (1% Triton X-100, 137 mM NaCl, 20 mM Tris pH 8.0, 0.15 U/mL aprotinin, 1 mM phenylmethylsulfonyl fluoride, 20 μM leupeptin; 0.5 mM sodium vanadate; 1 mL per 10 cm dish) at 4 °C for 15 min on a rocking platform. Lysates were collected and clarified of debris by centrifugation at 16,100×$g$ for 10 min at 4 °C.

To isolate the BRAF complexes, 5 mL of Halolink resin (Promega) was washed twice with Triton X-100 lysis buffer prior to the addition of lysate. The samples were then incubated for 2 h at 4 °C on a rocking platform. Beads containing the bound BRAF complexes were washed twice with Triton X-100 lysis buffer and three times with elution buffer (137 mM NaCl and 20 mM Tris pH 8.0). The bead-bound complexes were then resuspended in 2.5 mL elution buffer containing 50 µL Halo-TEV (Promega) and incubated for 2 h at 4 °C on a rocking platform. The beads were then pelleted, and the supernatant containing the eluted BRAF complexes was applied to a Superdex 6 Increase 10/300 GL column (Cytiva) pre-equilibrated with a buffer containing 20 mM Tris pH 8.0 and 137 mM NaCl. Proteins from the peak fractions corresponding to BRAF complexes were collected and analyzed by SDS–PAGE and silver staining.

**Cryo-EM grid preparation, data acquisition, and processing.** After gel filtration, the BRAF:14-3-3$_2$:MEK, BRAF:14-3-3$_2$, and BRAF$_2$:14-3-3$_2$ complexes (in 20 mM Tris pH 8.0 and 137 mM NaCl) were concentrated to 0.18–0.20 mg/mL by centrifugation in a 30K pore size Pall's Microsep™ advance centrifugal device. The samples were supplemented with NaCl and DTT to a final concentration of 200 and 10 mM, respectively, before freezing. Quantifoil Au R 1.2/1.3 holey carbon girds were glow-discharged for 45 s at 20 mA in an alcohol environment. A volume of 1.5 µL of protein solution was applied to each side of the grids, and the samples were vitrified in a FEI Vitrobot Mark IV, using a force of 10 a.u. and a blotting time of 1.5–2.0 s at 4 °C with humidity >85%. The samples were then frozen by plunge-freezing the grids into liquid ethane cooled to approximately −180 °C by liquid nitrogen. 50 frames per movie were collected from the frozen-hydrated samples at a nominal magnification of ×130,000 for the BRAF:14-3-3$_2$:MEK sample (corresponding to 1.058 Å per physical pixel) using a Titan Kiros electron microscope (FEI) at 300 kV with a K2 summit direct detection camera (Gatan) in super-resolution mode. The slit width of the energy filter was set to 20 eV. Then, using SerialEM[36] data collection software, the micrographs were dose-fractionated into 50 frames with a total exposure time of 8 seconds and a total electron exposure of 57 electrons per Å$^2$, with defocus values ranging from −0.8 to −2.5 µM.

All cryo-EM data analysis was done using RELION 3.1[37]. For the BRAF:14-3-3$_2$:MEK complex, a total of 3976 micrographs were collected. The raw movies were aligned and gain corrected to compensate for sample movement and drift by MotionCor2[38] with 5 × 5 patches and a B-factor of 150. The Gctf program[39] was used to estimate the contrast transfer function (CTF) parameters of each motion-corrected image. The micrographs were screened to remove low-quality images and those with unqualified CTF power spectra. An initial set of particles were manually picked and used to generate 2D class templates for automatic particle picking, using a box size of 180 pixels. Particle selection was inspected to remove any contaminants and to add missed particles. The resulting particles were subjected to three rounds of 2D classification to identify and discard false positives or other apparent contaminants. Following 2D classification, particles were further selected, re-centered, and re-extracted for 3D classification. An initial model was generated and, based on structural integrity and map quality of the complex, the best 3D class was then used for 3D refinement. After 3D auto-refinement, Bayesian polishing and per-particle CTF refinements were applied, until the 3D refinement converged. The final map was sharpened using a B-factor of −100.5 Å$^2$, yielding a density map at a resolution of 3.7 Å, based on the gold standard FSC 0.143 criteria. Masked-based local refinement was also performed on the BRAF KD:MEK portion of the structure, which generated a density map with the same resolution and quality, and on the BRAF KD portion of the structure, which resulted in no further improvement of the map.

The monomeric BRAF:14-3-3$_2$ and dimeric BRAF$_2$:14-3-3$_2$ datasets were collected in a similar manner, with 50 frames per movie collected from the frozen-hydrated samples at a nominal magnification of ×105,000 (corresponding to 1.348 Å per physical pixel). The micrographs were dose-fractionated into 50 frames every 0.2 s with a dose rate of 5.5 e-/Å$^2$/s, a total exposure time of 10 s, and an accumulated dose of 55 electrons per Å$^2$. The datasets were further processed in the same manner as described for the BRAF:14-3-3$_2$:MEK dataset Supplementary Table 4.

Models for the autoinhibited BRAF complexes were built by the rigid-body fitting of the individual BRAF domains using PDB IDs 5J17 for the RBD[40], 1FAR for the CRD[41], 4MNE for the KD[34], and 3WIG[42] for MEK1 protomer, when present. Similarly, the active BRAF dimer complex was built by rigid-body fitting using PDB ID 2FB8[43] for the BRAF KD. For all structures, 14-3-3 was built by rigid-body fitting using 14-3-3ζ PDB ID 4FJ3[44]. Each subunit of 14-3-3 dimer was fit individually, and the BRAF pS365 and pS729 sites were built manually as were any additional residues needed. Fitting of the models into their respective maps was initially done using UCSF Chimera[45]. Manual adjustment of the model was performed in Coot[46], followed by iterative rounds of real-space refinement in Phenix[47] and manual fitting in Coot. Model validation was done using statistics from Ramachandran plots and MolProbity scores in Phenix and Coot. Statistics for the final refinements are shown in Supplementary Table 4. Figures were generated by UCSF Chimera. Structure deviations and electrostatic potential of surfaces were calculated using MatchMaker and Columbic Surface plugins, respectively, in UCSF Chimera.

Initial diagnostic data sets for the three BRAF complexes were collected on a Talos Arctica microscope (FEI) at 200 kV with a Gatan K3 Summit direct detection camera at either the NCI-Frederick cryo-EM Facility or at the John M. Cowley Center for High-Resolution Electron Microscopy, Arizona State University. Data sets for generating the structures with the final reported resolutions were collected on a Titan Krios microscope at 300 kV with a Gatan K2 Summit direct detection camera in super-resolution mode.

**Fluorescence polarization assay.** Fluorescence polarization assay was used to measure the binding of BRAF:14-3-3$_2$:MEK and BRAF:14-3-3$_2$ complexes to KRAS. GppNHp-loaded GFP-KRAS at concentration of 0.73 nM was mixed with 0.0057–2.4 µM of BRAF:14-3-3$_2$:MEK and BRAF:14-3-3$_2$ complexes in buffer containing 20 mM Tris pH 8.0 and 137 mM NaCl. Fluorescence polarization of the samples was measured in black flat-bottom assay plates (Corning) using a CLARIOstar microplate reader (BMG LABTECH) with 482 nm excitation and 540 nm emission. The data was analyzed and fitted to the anisotropy single association hyperbolic equation using Prism 8 software.

**NanoBRET RAF autoinhibition assay.** 293FT cells were seeded into six-well tissue culture plates at a concentration of 4 × 10$^5$ cells/well. 16 h after plating, cells were co-transfected using 5 ng of the indicated NanoLuc-RAF$^{CAT}$ construct and 20 ng of the indicated BRAF$^{REG}$-Halo construct. Twenty-four hours later, cells were collected and resuspended in serum-free/phenol-free Opti-MEM (Gibco). HaloTag 618 ligand was added to the cell suspension (1 µL/mL), and cells were seeded in quadruplicate into wells of a 384-well plate (BioTek) at a concentration of 8 × 10$^3$ cells per well, with the remaining cells plated into a fresh six-well tissue culture plate. Cells were incubated at 37 °C with 5% CO$_2$ for 24 h, following which NanoGlo substrate was added to each well of cells (10 µL/mL), and the donor (460 nm) and acceptor (618 nm) emissions were measured using a Perkins Elmer Envision plate reader (#2104-0010 A containing a 460 nm/50 nm emission filter and a 610 nm LP filter). Cells seeded into the six-well plates were lysed and examined by immunoblot analysis using antibodies recognizing the Halo-tag (for RAF$^{REG}$ detection) and NanoLuc-tag (for RAF$^{CAT}$ detection) to ensure equal expression levels.

**Transfection, cell lysis, and co-immunoprecipitation assays.** HeLa or 293FT cells were plated into 10 cm tissue culture dishes at a concentration of 0.5 × 10$^6$/dish, 18–24 h prior to transfection. Cells were then transfected using the XtremeGENE9 transfection reagent, per the manufacturer's instructions, at a 2:1 ratio of XtremeGENE9 to DNA. 30 h after transfection, cells were serum-starved for 18 h, prior to lysis. For cell lysis, cells were washed twice with ice-cold PBS and lysed in 1% NP-40 buffer (20 mM Tris pH 8.0, 137 mM NaCl, 10% glycerol, 1% NP-40 alternative, 0.15 U/mL aprotinin, 1 mM phenylmethylsulfonyl fluoride, 0.5 mM sodium vanadate, and 20 µM leupeptin) at 4 °C for 15 min on a rocking platform. Lysates were clarified by centrifugation at 16,100×g for 10 min at 4 °C, following which the protein content was determined by Bradford assays. Lysates containing equivalent amounts of protein were incubated with the appropriate antibody and protein G sepharose beads for 2 h at 4 °C on a rocking platform. Complexes were washed extensively with 1% NP-40 buffer and then examined by immunoblot analysis, together with aliquots of equalized total cell lysate.

**Focus forming assay.** Recombinant retroviruses expressing the RAF proteins of interest were generated by transfecting 6 µg of the indicated pBabe DNA construct into a 100 mm tissue culture dish of Phoenix-Eco cells using the XtremeGENE9 protocol described above. Viral supernatants were collected 48 h post-transfection, centrifuged twice at 526×g for 10 min, and either stored at −80 °C or used directly. NIH-3T3 cells were plated into 60 mm tissue culture dishes at a concentration of 2 × 10$^5$ cells/dish. After 16 h, cells were infected with the indicated recombinant retrovirus in media containing 4% FBS and 8 µg/mL polybrene. 24 h later, cells were trypsinized and plated into two 100 mm dishes, one of which contained 5 µg/mL puromycin. After 2–4 weeks of culture, cells were fixed with 3.7% formaldehyde and stained with 1% methylene blue.

**Pull-down assays using Halo-RAS or Halo-RBD beads.** For bead preparation, 293FT cells transiently expressing the indicated Halo-KRAS$^{G12V}$ or Halo-RBD constructs were lysed in RIPA lysis buffer (20 mM Tris pH 8.0, 137 mM NaCl, 10% glycerol, 1% NP-40 alternative, 1% SDS, 0.1% sodium deoxycholate, 0.15 U/mL aprotinin, 1 mM phenylmethylsulfonyl fluoride, 0.5 mM sodium vanadate, and 20 mM leupeptin; 1 mL per 10 cm dish) at 4 °C for 15 min on a rocking platform. Lysates were clarified by centrifugation at 16,100×g for 10 min at 4 °C. Halolink resin (Promega) was washed twice with RIPA buffer and then added to the lysate (30 µL of bead resin per mL of lysate). Samples were incubated at 4 °C for 2 h on a rocking platform. Beads containing the Halo-tagged proteins were then washed once with RIPA buffer and twice with Triton lysis buffer (1% Triton, 137 mM NaCl, 20 mM Tris pH 8.0), prior to resuspension in Triton X-100 lysis buffer.

For pull-down assays using cell lysates, 293FT cells expressing the desired RAS or BRAF proteins were lysed in Triton X-100 lysis buffer, following which the protein content of the lysates was determined by Bradford assays. Lysates containing equivalent amounts of protein were incubated with the Halo-RAS or Halo-RBD beads (30 µL beads per 1 mL lysate) at 4 °C for 1.5 h on a rocking

platform. The beads containing the pull-down complexes were washed once with Triton lysis buffer and twice with elution buffer (137 mM NaCl and 20 mM Tris pH 8). The bead pellet was resuspended in 150 μL elution buffer containing 2 μL Halo-TEV (Promega) and incubated at 4 °C for 30 min. Beads were then pelleted and the eluate was collected for immunoblot analysis. For pull-down experiments using the purified BRAF:14-3-3$_2$ and BRAF:14-3-3$_2$:MEK complexes, 500 μL of the gel filtration fractions containing these complexes was incubated with 30 μL of washed KRAS$^{G12V}$ beads at 4 °C for 1.5 h on a rocking platform. Beads were then washed three times with elution buffer, and the eluate was collected as described above.

**BRET RAS–RAF interaction assay**. 293FT cells were seeded into 12-well dishes at a concentration of $1 \times 10^5$ cells/well. 16 h after plating, Venus-tagged and Rluc8-tagged constructs were co-transfected into cells using a calcium phosphate protocol. A duplicate 12-point saturation curve was generated in which the concentration of the energy donor construct (Rluc8) was held constant (62.5 ng) as the concentration of the energy acceptor plasmid (Venus) increased (0–1.0 μg). Cells were collected 48 h after transfection, washed, and resuspended in PBS (500 μL). 30 μL of the cell suspension was plated in duplicate into wells of a 384-well white-walled plate (PerkinElmer CulturPlate) and coelenterazine-h was added to a final concentration of 3.375 μM. The BRET and Rluc8 emissions were measured simultaneously using a PHERAstar Plus plate reader (BMG Labtech), with BRET monitored at 535 nm (bandwidth 30 nm) and Rluc8 measured at 475 nm (bandwidth 30 nm). To monitor the increasing levels of acceptor expression, 90 μL of the cell suspension was also plated in duplicate into wells of a 96-well black-walled plate (PerkinElmer OptiPlate), and Venus fluorescence was determined using an excitation wavelength of 485 nm (5 nm bandwidth) and the emission monitored at 530 nm (5 nm bandwidth) using a Tecan Infinite M1000 plate reader. The BRET ratio was determined by calculating the 535/475 ratio for a given sample (sample BRET) and was normalized to the BRET signal from cells expressing the donor construct alone to enable comparisons between experiments. The normalized mBRET signal was calculated as follows: $(1000 \times (\text{sample BRET ratio/background BRET ratio})) - 1000$. The acceptor/donor ratio (Venus emission/Rluc8 emission) for each data point was background corrected and equalized against a control where equal quantities of the Venus (acceptor) and Rluc8 (donor) constructs were transfected to allow for comparisons between experiments. Data were analyzed using GraphPad Prism. Non-linear regression was used to plot the best fit hyperbolic curve, and values for BRET$_{max}$ and BRET$_{50}$ were obtained from the calculated best-fit curves.

**Reporting summary**. Further information on research design is available in the Nature Research Reporting Summary linked to this article.

## Data availability

Three-dimensional cryo-EM density maps have been deposited in the Electron Microscopy Data Bank under accession numbers EMD-23813 (BRAF:14-3-3$_2$:MEK), EMD-23814 (BRAF:14-3-3$_2$), and EMD-23815 (BRAF$_2$:14-3-3$_2$). The coordinates of atomic models have been deposited in the Protein Data Bank under accession numbers PDB ID: 7MFD (BRAF:14-3-3$_2$:MEK), 7MFE (BRAF:14-3-3$_2$), and 7MFF (BRAF$_2$:14-3-3$_2$). Previously published accession number PDB IDs used in this manuscript are: 1FAR, 2FB8, 3WIG, 4FJ3, 4MNE, 4RZV, 5J17, 6NYB, 6UAN, 6U2G, 6U2H, 6XI7, 7JHP. Source data are provided with this paper.

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

## Acknowledgements

This research was supported by Federal funds from the National Cancer Institute, National Institutes of Health, under project numbers ZIA BC 010329 (D.K.M.) and ZIA BC 011744 (P.Z.). This work utilized the NCI/NICE cryo-EM Facility, and we would like to thank Dr. Rick Huang for help with cryo-EM data collection. The authors also thank Dr. Dan Shi and Dr. Dwight Williams for their assistance in collecting diagnostic cryo-EM data at the NCI-Frederick cryo-EM Facility and at the John M. Cowley Center for High-Resolution Electron Microscopy, Arizona State University, respectively and for their helpful discussions. In addition, we thank Dr. Ming Zhou (Inova Schar Cancer Institute) for mass spectrometry analysis of the BRAF complexes, Elizabeth Terrell for technical assistance in performing the RAS/RAF BRET assays, and Dr. Dominic Esposito (FNLCR Protein Expression Laboratory) for the KRAS protein used in fluorescent polarization assays. This study utilized the computational resources of the High-Performance Computing Biowulf cluster of the NIH (http://hpc.nih.gov).

## Author contributions

J.A.M.F., D.E.D., D.K.M., P.Z. conceived the project, interpreted the results, and wrote the paper. D.E.D. expressed and affinity-purified all BRAF complexes. J.A.M.F. further purified all BRAF complexes by size-exclusion chromatography and performed fluorescence polarization assays. D.E.D. performed mutagenesis, co-immunoprecipitation, pull-down, and BRET assays. J.A.M.F. screened, optimized, prepared cryo-grid samples, and collected electron microscopy data. J.A.M.F. and P.Z. processed, built, and analyzed Cryo-EM models. D.K.M. and P.Z. provided funding resources and supervision.

## Funding

## Competing interests

The authors declare no competing interests.
