## [Peer Review File · Nature Communications]

REVIEWER COMMENTS

Reviewer #1 (Remarks to the Author):

In this manuscript, new important structural and biochemical insights about RAF/RAS/MEK/ERK signaling cascade are presented. The cascade is extremely important for cell growth and differentiation, and the development of cancer, RASopathy syndromes and other diseases, making it a natural target for various therapeutics. The authors embarked on clarifying important questions about the nature of BRAF complexes and the mechanism whereby signals are conveyed from RAS downstream to MEK and ERK. More specifically, the authors elucidated how BRAF transitions from an autoinhibited monomeric state to an activated dimeric state while engaging with activated RAS. Three cryo-EM structures of distinct BRAF complexes purified directly from human cells are reported. In two structures, a monomeric BRAF is bound to either a 14-3-3 dimer alone or a 14-3-3 dimer and MEK, and in the third one, a BRAF dimer is in complex with a 14-3-3 dimer and two small-molecule inhibitor (one per BRAF protomer). Surprisingly, the authors find that the RBD of BRAF is partially occluded by 14-3-3, necessitating structural rearrangements in order to permit RAS to bind to BRAF. The approximate location of the RBD was previously reported in Park et al (Nature, 2019) but the resolution was not sufficient to position the known RBD model and identify the clash with 14-3-3. In addition, the authors probed some of their propositions using a variety of binding and activity assays, and then thoroughly discussed their findings in the context of earlier studies, especially three structural reports on similar targets that have been published in last two years. Taken together, the manuscript is well written and organized and is worthy of publication in Nature Communications. However, before publishing, the authors need to address a few major and a list of minor concerns.

Major concerns:

1. The authors note that the nature of binding between RAS and BRAF RBD may be different on initial contact than the state captured in the existing crystal structures. However, the authors write on page 11 that “the initial contact with RAS is made by the RAF RBD, whereby the B2-strand and the α 1-helix of the RBD interacts with the B2-strand and the switch I region of RAS.” The interactions described here are the ones from the crystal structures of what the authors eventually characterize as potentially the “final” state, not the “initial” state. So, this sentence should be rephrased to somehow indicate that this is the state that has been crystallized, and not explicitly that it is the “initial” interaction.
2. Along these same lines, on page 16 the authors state that “the structures of autoinhibited BRAF and CRAF complexes may vary and/or the contacts between RAS and the RBD-CRD regions of BRAF and CRAF may be different.” If it is true that the contact between RAS and CRAF are different than the ones seen for BRAF, then the interpretation of the mutational analysis involving the MM to KV

described in earlier portions of the paper (and shown in Figure 4) may no longer be valid. The authors should make sure that all of the explanations/interpretations are consistent with the model they put forth, which will require acknowledging some of the uncertainties involved here.

3. Additionally, the authors write several times that there is “a conformational change in 14-3-3 binding”, and the wording here somewhat implies (and at the very least could be confusing to the reader) that there is a conformational change in 14-3-3. However, the 14-3-3 models align very well between the BRAF monomer and BRAF dimer states (in fact, the authors state that the same 14-3-3 model was used for initial fitting). Are the authors saying that these are two different conformations? It seems unlikely that 14-3-3 starts in one conformation (BRAF monomer), changes to a new conformation (upon RAS binding), and then goes back to the original conformation (BRAF dimer), and, in any event, there is no direct evidence for such a conformational change. Thus, a better way to characterize the change is just that the interaction between 14-3-3 and the RBD gets disrupted or shifted, due to the effects (steric clash and electrostatic repulsion) described by the authors. The wording “conformational change in 14-3-3 binding” could of course be interpreted as merely a rigid body movement, but as written, it could be confusing to the reader. So, if the authors intended to portray just a rigid body movement, they should state so more clearly.

Minor concerns:

- Figure 2C, part of the sequence is cut off on the left side (M, K, and K are cut off)
- Page 10, “reavealed” should be “revealed”
- Page 11, “auoinhibition” should be “autoinhibition”
- Page 11; last sentence: “individual” and “each of the four” should be removed
- Page 12, penultimate sentence: “in turn” before “the CRD” should be removed
- Page 13, line 7: “clash cause” should state “clash caused”
- Page 15, line 7: “may represent addition levels” should state “may represent an additional level”
- Page 15, lines 18-23; sentences should be rephrased perhaps into something like this: “Recent studies about interactions of CRAF RBD and CRAF RBD-CRD with KRAS indicated that residues in the analogous position in the CRAF RBD (K87/V88) make van der Waals contacts with I24, Q25 and Y40 in RAS. Although the structure of RAS in complex with BRAF RBD is lacking, our mutational analyses show that M186 and M187 may modulate binding of the BRAF RBD to KRAS.”
- Page 16, line 4: “in turn” before “the CRD” should be removed
- Page 17, line 17: replace “tease out” with “delineate/determine/define”

- Figure 3C: Would it not be more optimal to present KD values as nM?
- Figure 4B: Would it not be better to show these results as normalized values where WT is 1 (or 100)?
- Check methods section for typographical errors!
- Figure S1A-C: lines in chromatograms should be thicker and colored in darker shade.
- Figures S2D, S3D, and S4D: 3D maps are colored as (extremely) light grey and they are not visible. Please provide these panels with maps in a bit darker shade.

Reviewer #2 (Remarks to the Author):

“Structural insights into the BRAF Monomer-to-dimer Transition Mediated by RAS binding” by Martinez-Fiesco and Durrant et al.

The MAPK pathway is one of the most mutated in cancer and BRAF is the most mutated protein Kinase in all cancers. Despite decades of efforts to understand the nature of Inactive and Active RAF containing complexes, we have just recently begun to visualize the structures (Kuriyan, Eck, Sudhamsu labs) of these complexes that shed light and help us better understand the regulated of this pathway. The current study adds to the structural information from the past 2 years and helps us get closer to better understand how Ras-GTP activates RAF – currently uncharacterized in literature.

There are two main points that are advances compared to existing literature:

- (1) Structural characterization of the interaction of RAF RBD with 1433 in the inactive RAF complexes
- (2) Validation of the importance of that interface by mutational studies and impact on pathway signaling as well as structural analysis / incompatibility of Ras binding to 1433 bound RBD

Overall, in this manuscript, structures of 3 complexes have been determined, 2 of which reveal novel information. These are a first, for complexes purified from mammalian cells, rather than insect cell overexpression.

The resolution of RBD in the inactive RAF structure is an important new finding and the methods employed in the manuscript offer alternative approaches to generate inactive / active RAF complexes for further research (endogenous level expression in mammalian cells vs insect cell overexpression)

Overall, despite the novelty of the 2 main points I mentioned above, I do have several concerns regarding multiple interpretations / conclusions in the manuscript, which if addressed would improve the manuscript.

(a) The authors observe two different inactive RAF complexes – one with and one without MEK bound and do not observe electron density for ATP in the BRAF (or MEK) active site in their moderate-to-low resolution structures and conclude that neither MEK nor ATP is necessary for the inactive BRAF-1433 complex to form. The concentration of BRAF in human cells is ~10-100 nM range and the concentration of MEK is in the 2 micromolar range and concentration of ATP is ~1 mM. The affinity of MEK for BRAF is known to be ~40-50 nM (Haling, Cancer Cell – 2014) and the affinity of ATP for MEK is ~100 nM and the affinity of ATP for BRAF is ~20 micromolar as reported in literature. Affinity of 1433 for BRAF is in the picomolar range. Given the extensive purification steps that the authors employed, and the various known affinities, it is extremely conceivable and likely that some of the MEK, most of the ATP (bound to BRAF) was lost during purification and the two inactive complexes that the authors observe are a result of the purification process, where ~ 2 micromolar MEK, and 1 mM ATP in the cell enabled inactive RAF conformation (Park 2019, Liau 2020) formation of the inactive complex where, not co-expressing MEK resulted in a majority active RAF complex, and co-expression of MEK resulted in the inactive complex when wild type RAF was expressed.

(b) CryoEM electron density maps and interpretations: The authors suggest that the conformation of the BRAF kinase domain is identical between the two inactive complexes BRAF:(1433)₂:MEK and the BRAF:(1433)₂ - and to the published structures from the Eck and Sudhamsu labs - giving the impression that MEK (or ATP) has no effect on the inactive complex and that the residue level details of the BRAF active site are well defined in both structures. However, the map in Fig S3B shows that the loss of MEK leads to greatly increased flexibility or disorder of the BRAF kinase domain. The authors should analyze this part of the structure and perform some model building based on the electron density observed, as much as possible – or delete residues when they are not found to agree with the density. The N-lobe in particular looks to be highly flexible/poorly resolved and may be taking on multiple conformations. The authors clearly show that 14-3-3 binding to pS729 and pS365 is sufficient to lock BRAF in a monomeric state in the absence of MEK and ATP, yet, given

current debates in the field regarding the role of MEK on BRAF-containing complexes, a slightly more nuanced discussion which better represents the BRAF:(14-3-3)₂ structural data is recommended.

(c) The authors propose RBD interactions with 14-3-3 maintain the inactive BRAF complex, and that Ras binding to the RBD dislodges the RBD from 14-3-3 given steric clashes. This is proposed as the first step toward BRAF activation. Regarding RBD-14-3-3 interactions, given the middling effect in the BRAFREG:BRAF^{KD} BRET assay of point mutations to the RBD compared to the T241P point mutation in the CRD, seeing the effect of a more stringent alteration (e.g. mutation to negatively charged residues or deletion of the RBD), would help to better assess the importance of this interaction. The “dislodging” activity of Ras remains speculative (Ras binds tightly to both BRAF:(14-3-3)₂ complexes and the authors do not test altering residues contributing to the proposed Ras:RBD clash), but proving this model could constitute a new study beyond the scope of this manuscript.

(d) How do the authors compare their results with the Eck structure, where the RBD was found to be disordered, whereas it is ordered in this manuscript? The difference mentioned is that the sample in this case was from mammalian cells. Was this due to potential post-translational modifications? If so, what are they? Knowing that would be very important to rationalize. Was this due to differences in data collection or data processing OR a real biological difference? What is particular about the mammalian cell expression “more closely represents” authentic signaling complexes?

Specific comments:

(1) Page 5, line 24 and and Figure S1D: “...the three sample complexes revealed that 14-3-3 ϵ and 14-3-3 ζ constituted approximately 50% and 25%, respectively of the 14-3-3 proteins... all other 14-3-3 isoforms were detected.

Can the authors comment on whether the % of 14-3-3 isoforms in the complexes reflect whole cell % of isoforms? Or is BRAF preferentially forming complexes with 14-3-3 ϵ and 14-3-3 ζ ? If it is, are there any conclusions that one can draw from the structures and interacting residues as to why this may be?

(2) Page 6, line 15: “despite the presence of MEK in the cryo-EM sample fraction, no density corresponding to MEK was observed in the 3D reconstructions...” and Figure S1A-C.

Can the authors quantify relative band densities of BRAF, 14-3-3 and MEK in EM sample fractions in Fig. S1A-C? MEK appears sub-stoichiometric in all samples, but particularly reduced in the BRAF₂:14-

3-32 complex. Could the lack of a MEK inhibitor in the sample also explain missing MEK density when compared to results from Park et al (PDBID 6q0j) or Haling et al (PDBID 4mne)? In addition to cellular BRAF complexes not containing MEK, it also seems likely MEK could be dissociating from intact MEK:BRAF:14-3-3 complexes during the purification process given reported BRAF:MEK affinities (50-100 nM), whereas 14-3-3 stays bound due to avidity considerations (see comments above).

(3) Page 6, line 26: “our structure superimposing well onto the ligand-free dimer structure, with an overall C α root mean square deviation (RMSD) of 1.03 Å”

pg. 7, line 18 “overlap well with the high resolution crystal structure of inhibitor (GDC-0879)-bound BRAF KD (residues D432–R735):14-3-32 complexes (C α RMSD of 0.80 Å)” and Figure 1C.

Is the RMSD specific to the BRAFKD dimer, or the overall BRAF2:14-3-32 complex? In Fig 1C it would be useful to align the BRAF2:14-3-32 complex reported in this manuscript against the Kondo et al structure (6uan) using only the 14-3-3 dimer or only the BRAF dimer. It appears the pose of the BRAF dimer relative to 14-3-3 differs between those two structures? To that end, it would be useful to align the current structure against the much higher resolution Liau et al crystal structure (6u2h or 6xag). Do the two EM structures and the crystal system represent different juxtapositions of 14-3-3 dimer vs. BRAF dimer? In Fig 1C, can the authors more clearly label the grey structure as 6UAN vs. current structure? 6UAN is difficult to make out with current coloring.

(4) Page 8, line 4: “Strikingly, except for the presence or absence of MEK, the overall conformation of the monomeric BRAF structures was nearly identical...”

line 11 “...they demonstrate that MEK binding is not required for BRAF to assume a stable, autoinhibited conformation. The BRAF kinase domain in both our monomer structures displayed the canonical kinase inactive conformation...”

line 17 “the structure of the KDs displayed an almost identical conformation as did recent inactive KD structures bound to ATP analogs...”

line 23 “These findings indicate that the ATP binding pocket of the BRAF KD is stable in the “apo” nucleotide-free state”.

P. 14, line 22 “our structures show that the ATP-binding site is fully formed in the “apo” state and that stabilization of the autoinhibited state is also not dependent on ATP binding” Fig. 2C, and Fig S3B

The authors comment multiple times that except for MEK occupancy the BRAF:(14-3-3)2:MEK and BRAF:(14-3-3)2 structures are identical, including secondary structure details of the BRAF ATP-binding pocket. Figure 2A only shows the BRAF:(14-3-3)2:MEK map, implying a similar map for

BRAF:(14-3-3)₂. However, the map presented for BRAF:(14-3-3)₂ in Fig. S3B shows significantly poor density for BRAFKD, and the N-lobe in particular looks to be highly flexible/poorly resolved and is likely due to the N-lobe taking on multiple conformations. This suggests MEK and ATP do influence BRAF conformation in the inactive complex and this map ought to be included in the main text figures. Can the authors show and compare local EM density between both +/- MEK structures for the BRAF and MEK ATP binding pocket, including key elements such as the P-loop (res 463-471 in BRAF) and inhibitory turn (res 598-602) to support claims that ATP binding has no effect on this conformation for either structure? Can the authors show both a BRAFKD protomer in the active (i.e. PDB ID 2fb8, 4mnf or similar) and inactive conformation (i.e. 6pp9 or 6u2g) fit into density for the BRAF-Clobe in the BRAF:(14-3-3)₂ map to better assess the conformation taken on by the BRAF-N-lobe in this map?

(5) p. 9, line 17: "A distinct feature of the BRAF structures reported here is the resolution of the RBD in the context of the full-length, autoinhibited BRAF monomer" line 23: "an extensive contact surface of ~435 Å² is observed between the RBD and the 14-3-3 protomer", Fig 2C:

Can the authors comment/speculate as to why the RBD and this interface were not resolved in Park et al structure, but have clear density in the maps reported here? Are the key residues conserved in insect 14-3-3? Can the authors overlay semi-transparent map density in Fig 2C to help the reader evaluate this interface? Is there side-chain density for M186 and M187, given that the overall resolution of the reconstruction appears to be best around this RBD/1433 region?

(6) p. 10, line 21 "a reproducible 10% decrease in signal was observed for the M186W/M187W mutant, whereas a 20% increase in signal was observed for the M186A/M187A mutant" Fig 2D:

Do the authors believe a 10% reduction in BRET signal represents the full effect of ablating the RBD/14-3-3 interface, as compared to 50% reduction seen for the T241P CRD mutant? Given subtle effects of M186/M187 mutants, it would be helpful to see the effect of harsher mutations (e.g. to negatively charged residues) or deletion of the RBD in the BRET assay to confirm the full effect of breaking the RBD/14-3-3 on the BRAF inactive complex.

(7) p. 11, line 19 "both complexes exhibited a high degree of binding and had affinities in the nanomolar range." and Fig. 3C

Can the authors comment on how the measured Ras affinities to the BRAF:14-3-3:MEK and BRAF:14-3-3 complexes compare to Ras binding to the BRAF-RBD alone? Does presence of the full complex

increase, decrease or have no effect on Ras binding? On that end, Does the RBD bind to 1433 directly with reasonable affinity (even if it is micromolar?) – This could be tested directly.

(8) p. 12 line 7: “Specifically, RAS residues I21, Q22 (α 1-helix), Q25-E31 (SI region), K42 and V45 (β 2 sheet) would clash with 14-3-3... this region of 14-3-3 has a notable negative charge and, while complementary with the positively charged RBD interface, would cause electrostatic repulsion with negatively charged residues in KRAS switch I... residues D30 and E31 would be brought in close proximity to D197 and E198 of 14-3-3.”

This structural analysis predicts point mutants to Ras relieving steric clashes and electrostatic repulsion would improve Ras binding to inactive RAF complexes. If proved true, this would better support speculation that Ras acts in part by dislodging the RBD from 14-3-3.

(9) Minor points: Page 5 – SB 590885 is a Type 1 inhibitor; Figure 2F – It is unclear if the protein expression levels are equal for the bottom two plates. Please control for this as the cause of the differences in colony counts; Figure 4D: It may be better to show all the figures with the same perspective. It is difficult to compare the four panels when the model moves only slightly between each of them.

Finally, the title is suggestive of more than what is presented in the manuscript. The monomer to dimer transition for RAF involves much more than Ras binding. The SHOC2-pp1c-Ras complex needs to de-phosphorylate the pS-365 (BRAF), two RAF molecules need to come together and one 1433 dimer molecule needs to be displaced and set free. How all of that occurs is yet to be discovered. It is possible to conceive other intermediate steps. So, the title could be modified to better reflect the content of the manuscript.

Overall, the manuscript contains data that could be considered for publication, given the authors address the concerns / comments listed above.

Reviewer #3 (Remarks to the Author):

In this manuscript by Martinez Fiesco, Durrant and colleagues, the group reports on several structures of BRAF isolated from mammalian cells. This includes cryoEM structures of a BRAF dimer bound to 14-3-3 isolated in complex with the inhibitor SB590885, as well as structures of the BRAF monomer bound to 14-3-3 but in the presence and absence of MEK. The structures show similar features to recently reported cryoEM structures reported by Park et al., however, the authors in this manuscript also reveal the position of the RAS-binding domain (RBD), which was not resolved or previously reported in other structures. The insights on the RBD and 14-3-3 interactions are highly novel and allow the authors to propose a model for how the monomer to dimer transition in BRAF is mediated by RAS binding. The manuscript is well written and the implications of the structures to our understanding of RAS-RAF signaling are supported through mutational and functional studies, making this a highly compelling body of work. I have listed a few points below for the authors - the request of additional mutations within the RBD could be considered but are not essential.

Specific comments.

1. Can the authors mutate the interface residues M186/M187 to negatively charged amino acids (eg. MM>EE)? It would be interesting to see how such mutations compare in the assays shown in Figure 2D-F. Based on their model, such mutations may disfavor binding to 14-3-3 but also RAS binding.
2. Can the authors add additional panels or images to clarify Figure 3F? The arguments and novelty in this section are potentially very high but the images could be improved to better illustrate specifically the repulsion between RAS and 14-3-3 that could facilitate rearrangement of BRAF to dislodge the auto-inhibited complex.
3. Part of the alignment in Figure 2C appears to be cut-off.
4. In Figure S3D, can the authors add images of the entire RBD showing similar model and map overlays; several orientations can be shown. Such images would be complementary to the individual segments already shown in this panel but would give a better sense of map quality and fit throughout this domain.

Response to the Reviewers' Comments:

We would like to thank the reviewers for their supportive comments and helpful suggestions. We have addressed their comments and concerns as outlined below (in blue). The revised manuscript now contains new data including the evaluation of a BRAF M186E/M187E mutant and the further analysis of the monomeric BRAF:14-3-3₂ and BRAF:14-3-3₂:MEK structures. In addition, we have revised the text to incorporate this new data and to address the reviewers specific comments.

Reviewer #1 (Remarks to the Author):

In this manuscript, new important structural and biochemical insights about RAF/RAS/MEK/ERK signaling cascade are presented. The cascade is extremely important for cell growth and differentiation, and the development of cancer, RASopathy syndromes and other diseases, making it a natural target for various therapeutics. The authors embarked on clarifying important questions about the nature of BRAF complexes and the mechanism whereby signals are conveyed from RAS downstream to MEK and ERK. More specifically, the authors elucidated how BRAF transitions from an autoinhibited monomeric state to an activated dimeric state while engaging with activated RAS. Three cryo-EM structures of distinct BRAF complexes purified directly from human cells are reported. In two structures, a monomeric BRAF is bound to either a 14-3-3 dimer alone or a 14-3-3 dimer and MEK, and in the third one, a BRAF dimer is in complex with a 14-3-3 dimer and two small-molecule inhibitor (one per BRAF protomer). Surprisingly, the authors find that the RBD of BRAF is partially occluded by 14-3-3, necessitating structural rearrangements in order to permit RAS to bind to BRAF. The approximate location of the RBD was previously reported in Park et al (Nature, 2019) but the resolution was not sufficient to position the known RBD model and identify the clash with 14-3-3. In addition, the authors probed some of their propositions using a variety of binding and activity assays, and then thoroughly discussed their findings in the context of earlier studies, especially three structural reports on similar targets that have been published in last two years. Taken together, the manuscript is well written and organized and is worthy of publication in Nature Communications. However, before publishing, the authors need to address a few major and a list of minor concerns.

RESPONSE:

We greatly appreciate the reviewer's positive comments.

Major concerns:

1. The authors note that the nature of binding between RAS and BRAF RBD may be different on initial contact than the state captured in the existing crystal structures. However, the authors write on page 11 that "the initial contact with RAS is made by the RAF RBD, whereby the B2-strand and the α 1-helix of the RBD interacts with the B2-strand and the switch I region of RAS." The interactions described here are the ones from the crystal structures of what the authors eventually characterize as potentially the "final" state, not the "initial" state. So, this sentence should be rephrased to somehow indicate that this is the state that has been crystallized, and not explicitly that it is the "initial" interaction.

RESPONSE:

We agree with the reviewer's point and have rewritten this sentence. The revised sentence now states "Previously determined crystal structures of RAS-RBD complexes have shown that the β 2-strand and the α 1-helix of the RBD interacts with the β 2-strand and the switch I region of RAS^{35,36} and that RAS-RBD binding involves ionic and hydrogen bonds as well as other Van der Waals interactions³⁷."

2. Along these same lines, on page 16 the authors state that “the structures of autoinhibited BRAF and CRAF complexes may vary and/or the contacts between RAS and the RBD-CRD regions of BRAF and CRAF may be different.” If it is true that the contact between RAS and CRAF are different than the ones seen for BRAF, then the interpretation of the mutational analysis involving the MM to KV described in earlier portions of the paper (and shown in Figure 4) may no longer be valid. The authors should make sure that all of the explanations/interpretations are consistent with the model they put forth, which will require acknowledging some of the uncertainties involved here.

RESPONSE:

We thank the reviewer for pointing out the potential confusion the indicated sentence could cause. We have removed this sentence from the revised text and now end the paragraph on page 16 with the following statement “Thus, the RAS:CRAF-RBD-CRD structure may be more representative of the final RAS-BRAF binding conformation rather than the initial contact between RAS and the autoinhibited BRAF complexes.”

3. Additionally, the authors write several times that there is “a conformational change in 14-3-3 binding”, and the wording here somewhat implies (and at the very least could be confusing to the reader) that there is a conformational change in 14-3-3. However, the 14-3-3 models align very well between the BRAF monomer and BRAF dimer states (in fact, the authors state that the same 14-3-3 model was used for initial fitting). Are the authors saying that these are two different conformations? It seems unlikely that 14-3-3 starts in one conformation (BRAF monomer), changes to a new conformation (upon RAS binding), and then goes back to the original conformation (BRAF dimer), and, in any event, there is no direct evidence for such a conformational change. Thus, a better way to characterize the change is just that the interaction between 14-3-3 and the RBD gets disrupted or shifted, due to the effects (steric clash and electrostatic repulsion) described by the authors. The wording “conformational change in 14-3-3 binding” could of course be interpreted as merely a rigid body movement, but as written, it could be confusing to the reader. So, if the authors intended to portray just a rigid body movement, they should state so more clearly.

RESPONSE:

We appreciate the reviewer’s comment and agree that using the phrasing “a conformational change in 14-3-3 binding” is confusing as we did not mean to imply “a conformational change within the 14-3-3 dimer”. In the revised text, the section describing our model has been rewritten, and we now state that “Formation of the ionic bonds with RAS would generate steric clashes and electrostatic repulsion at the RBD:14-3-3 interface that would initiate a rearrangement in 14-3-3 dimer binding, resulting in the exposure of additional RBD residues involved in full RAS contact.”

Minor concerns:

- Figure 2C, part of the sequence is cut off on the left side (M, K, and K are cut off)

RESPONSE:

The full sequence is now shown.

- Page 10, “reavealed” should be “revealed”

RESPONSE:

This misspelling has been corrected.

- Page 11, “auoinhibition” should be “autoinhibition”

RESPONSE:

This misspelling has been corrected.

- Page 11; last sentence: “individual” and “each of the four” should be removed

RESPONSE:

The original sentence had included these words to prevent the reader from thinking that only one mutant had been generated, which contained substitutions in all four basic residues. After reading the reviewer’s comment, we realize that this phrasing is wordy and may still be confusing to a reader. In the revised manuscript, we have changed the sentence to read: “As expected, mutation of any of the RBD basic residues that form ionic bonds with KRAS (R158A, R166A, K183A, or R188L), significantly disrupted KRAS^{G12V} binding in the pull-down assays (Fig. 4e), confirming that RBD contact is essential for the RAS-BRAF interaction.”

- Page 12, penultimate sentence: “in turn” before “the CRD” should be removed

RESPONSE:

“in turn” has been removed and the sentence has been rewritten to state “We predict that the full spectrum of RAS-RBD interactions would dislodge the RBD from the autoinhibited complex as well as the CRD, due to the short linker between these two domains, thereby enabling the CRD to rotate and make contact with RAS and the plasma membrane”.

- Page 13, line 7: “clash cause” should state “clash caused”

RESPONSE:

This correction has been made.

- Page 15, line 7: “may represent addition levels” should state “may represent an additional level”

RESPONSE:

This correction has been made.

- Page 15, lines 18-23; sentences should be rephrased perhaps into something like this: “Recent studies about interactions of CRAF RBD and CRAF RBD-CRD with KRAS indicated that residues in the analogous position in the CRAF RBD (K87/V88) make van der Waals contacts with I24, Q25 and Y40 in RAS. Although the structure of RAS in complex with BRAF RBD is lacking, our mutational analyses show that M186 and M187 may modulate binding of the BRAF RBD to KRAS.”

RESPONSE:

These sentences were revised based on the reviewer’s recommendation and now read:

“Recent crystal structures of CRAF RBD and CRAF RBD-CRD in complex to KRAS indicate that residues in the analogous position in the CRAF RBD (K87/V88) make van der Waals contacts with residues I24/Q25/Y40 in RAS. Although the structure of RAS in complex with BRAF RBD is lacking, our mutational analysis of the BRAF RBD M186/M187 residues suggests that alterations in these residues can modulate KRAS binding”.

- Page 16, line 4: “in turn” before “the CRD” should be removed

RESPONSE:

We prefer to maintain this phrasing to emphasize that in our model the CRD is dislodged as a result of the RBD being dislodged.

- Page 17, line 17: replace “tease out” with “delineate/determine/define”

RESPONSE:

In accordance with the reviewer’s comment, we have replaced “Tease out” with “determine”.

- Figure 3C: Would it not be more optimal to present KD values as nM?

RESPONSE:

The KD values are now represented as nM.

- Figure 4B: Would it not be better to show these results as normalized values where WT is 1 (or 100)?

RESPONSE:

For the BRET saturations curves, it is standard in the field to show each of the data-derived curves from which the BRET₅₀ is derived (Lavoie *et al.*, 2013; Jin *et al.*, 2017; and Bery *et al.*, 2018). In addition, presenting the BRET saturation curves for these mutants allows the reader to compare these particular BRAF mutants with the analysis of other BRAF mutants that have been published in literature (Terrell *et al.*, 2019).

- Check methods section for typographical errors!

RESPONSE:

The methods section as well as other sections of the manuscript have been rechecked for typographical errors and corrected accordingly.

- Figure S1A-C: lines in chromatograms should be thicker and colored in darker shade.

RESPONSE:

We have changed the chromatograms as suggested.

- Figures S2D, S3D, and S4D: 3D maps are colored as (extremely) light grey and they are not visible. Please provide these panels with maps in a bit darker shade.

RESPONSE:

The coloring of the 3D maps has been changed as requested.

Reviewer #2 (Remarks to the Author):

“Structural insights into the BRAF Monomer-to-dimer Transition Mediated by RAS binding” by Martinez-Fiesco and Durrant et al.

The MAPK pathway is one of the most mutated in cancer and BRAF is the most mutated protein Kinase in all cancers. Despite decades of efforts to understand the nature of Inactive and Active RAF containing complexes, we have just recently begun to visualize the structures (Kuriyan, Eck, Sudhamsu labs) of these complexes that shed light and help us better understand the regulated of this pathway. The current study adds to the structural information from the past 2 years and helps us get closer to better understand how Ras-GTP activates RAF – currently uncharacterized in literature.

There are two main points that are advances compared to existing literature:

- (1) Structural characterization of the interaction of RAF RBD with 1433 in the inactive RAF complexes
- (2) Validation of the importance of that interface by mutational studies and impact on pathway signaling as well as structural analysis / incompatibility of Ras binding to 1433 bound RBD

Overall, in this manuscript, structures of 3 complexes have been determined, 2 of which reveal novel information. These are a first, for complexes purified from mammalian cells, rather than insect cell overexpression.

The resolution of RBD in the inactive RAF structure is an important new finding and the methods employed in the manuscript offer alternative approaches to generate inactive / active RAF complexes for further research (endogenous level expression in mammalian cells vs insect cell overexpression)

Overall, despite the novelty of the 2 main points I mentioned above, I do have several concerns regarding multiple interpretations / conclusions in the manuscript, which if addressed would improve the manuscript.

(a) The authors observe two different inactive RAF complexes – one with and one without MEK bound and do not observe electron density for ATP in the BRAF (or MEK) active site in their moderate-to-low resolution structures and conclude that neither MEK nor ATP is necessary for the inactive BRAF-1433 complex to form. The concentration of BRAF in human cells is ~10-100 nM range and the concentration of MEK is in the 2 micromolar range and concentration of ATP is ~1 mM. The affinity of MEK for BRAF is known to be ~40-50 nM (Haling, Cancer Cell – 2014) and the affinity of ATP for MEK is ~100 nM and the affinity of ATP for BRAF is ~20 micromolar as reported in literature. Affinity of 1433 for BRAF is in the picomolar range. Given the extensive purification steps that the authors employed, and the various known affinities, it is extremely conceivable and likely that some of the MEK, most of the ATP (bound to BRAF) was lost during purification and the two inactive complexes that the authors observe are a result of the purification process, where ~ 2 micromolar MEK, and 1 mM ATP in the cell enabled inactive RAF conformation (Park 2019, Liao 2020) formation of the inactive complex where, not co-expressing MEK resulted in a majority active RAF complex, and co-expression of MEK resulted in the inactive complex when wild type RAF was expressed.

RESPONSE:

We appreciate the reviewer's concern and realize that we may not have been clear in stating our points and conclusions. In the text, we did not mean to imply that "neither MEK or ATP is necessary for the inactive BRAF:14-3-3 to form", only that their presence is not required for BRAF to maintain the autoinhibited conformation. We realize that some of our word usage may have been confusing, and we have rewritten portions of the text to clarify these points.

With regard to ATP binding, we acknowledge that ATP is found at high concentrations in mammalian cells and that ATP binding may be needed for the formation of the inactive conformation. Moreover, we agree that ATP may have dissociated from the BRAF complexes during the purification process, resulting in the capture of the kinase domains in the 'apo' state. Nonetheless, our structures indicate that once the inactive, autoinhibited state is achieved, the conformation can be maintained in the absence of ATP binding, thus allowing cryo-EM structures to be obtained that are strikingly similar to inactive, BRAF kinase domain structures obtained using ATP analogues. In the text of the revised results section, we state that "As was reported for the ATP-analog-bound structures, the N- and C-lobes of the KD in our ATP-free structure exhibited a closer orientation than is observed for the lobes of KDs bound to RAF inhibitors" and that "These findings indicate the ATP binding pocket of the BRAF KD is stable in the "apo" nucleotide-free state and that while ATP-binding may be required to form the compact configuration of the N- and C-lobes as well as the autoinhibited BRAF conformation, these states can exist in the absence of bound ATP (Supplementary Fig. 5b-c).

With regard to MEK binding, we agree that in human cells the concentration of MEK exceeds the K_D value for binding to BRAF; therefore, BRAF-bound to MEK would be expected if binding affinity were the only factor considered. However, it is known that the translation and processing of the RAF kinases is a dynamic process and involves chaperone complexes (Hsp90/cdc37) as well as changes in phosphorylation states, which could also influence BRAF interactions. Our BRAF complexes were obtained from human cells stably expressing Halo-BRAF, such that complex formation could occur under conditions where normal cellular regulatory mechanisms would be intact. In all our preparations of the “inactive BRAF complexes” from serum-starved cells, we found that regardless of whether the cells were treated with the MEK inhibitor CH5126766 to stabilize the MEK:BRAF interaction, we consistently (and repeatedly) obtained a population of the BRAF:14-3-3₂ complex that was equivalent to the population of BRAF:14-3-3₂:MEK complex, as indicated in the gel chromatography fractions shown in Supplementary Fig. 1a, b. If MEK was simply dissociating from the complex during purification, one would expect that treatment with CH5126766, which stabilizes the MEK:BRAF interaction, would increase the proportion of MEK-bound complexes, but we found that CH5126766 treatment had little to no effect on the proportionality of the complexes. Moreover, the purification of our autoinhibited BRAF complexes was quite straightforward and involved affinity capture of the complexes from cell lysates, cleavage of the complexes from the affinity resin, and separation of the samples via gel chromatography, which was achieved in approximately 5 hrs., with the samples being maintained at 4°C. Therefore, we do not think that the two autoinhibited BRAF complexes (+/- MEK) simply results from the dissociation of MEK during purification but rather is indicative of distinct cellular pools of autoinhibited BRAF complexes, one that contains MEK and one that is MEK free. Although an in-depth characterization of the cellular BRAF pools and their regulation is beyond the scope of this study, our data are consistent with a model that the MEK-free complexes are more flexible (as indicated by the local resolution maps) and may be a precursor to the MEK-bound complexes, with MEK binding further stabilizing the autoinhibited state.

Finally, the reviewer indicated that when BRAF is not co-expressed with MEK, others have found that the majority of the BRAF complexes will be in the “active” dimeric state (Park *et al.*, 2019 and Liao *et al.*, 2020). However, this was not the case in our study, as is demonstrated in the gel filtration fractions shown in Supplementary Fig. 1b-c. More specifically, in our stable 293FT expression system and in the absence of MEK overexpression, the majority of the 14-3-3-containing BRAF complexes isolated from quiescent, serum-depleted cells were found to elute in fractions 18 and 19, not fraction 16, where our active, BRAF₂:14-3-3₂ complexes eluted. BRAF complexes in fractions 18 and 19 both yielded cryo-EM structures that were in the monomeric, autoinhibited state described by Park *et al.*, 2019, with critical elements of the kinase domain in the inactive conformation. (Also, see response to point (b) and (4) below).

(b) CryoEM electron density maps and interpretations: The authors suggest that the conformation of the BRAF kinase domain is identical between the two inactive complexes BRAF:(1433)₂:MEK and the BRAF:(1433)₂ - and to the published structures from the Eck and Sudhamsu labs - giving the impression that MEK (or ATP) has no effect on the inactive complex and that the residue level details of the BRAF active site are well defined in both structures. However, the map in Fig S3B shows that the loss of MEK leads to greatly increased flexibility or disorder of the BRAF kinase domain. The authors should analyze this part of the structure and perform some model building based on the electron density observed, as much as possible – or delete residues when they are not found to agree with the density. The N-lobe in particular looks to be highly flexible/poorly resolved and may be taking on multiple conformations. The authors clearly show that 14-3-3 binding to pS729 and pS365 is sufficient to lock BRAF in a monomeric state in the, absence of MEK and ATP, yet, given current debates in the field regarding the role of MEK

on BRAF-containing complexes, a slightly more nuanced discussion which better represents the BRAF:(14-3-3)₂ structural data is recommended.

RESPONSE:

We agree with the reviewer that the BRAF:14-3-3₂ complex appears to have increased flexibility, particularly in the kinase domain N-lobe (as indicated in the local resolution maps), resulting in an overall resolution of 4.1 Å compared to the 3.7 Å resolution of the BRAF:14-3-3₂:MEK complex. However, it is unclear whether this difference is due solely to the lack of MEK binding. Moreover, as described below, although this structure may have increased flexibility, we can clearly discern that it is in the “inactive KD conformation”. We would also like to note that we now realize that Supplementary Fig. 3B contained an earlier, suboptimal density map. Revised Supplementary Fig. 3b now contains the appropriate display of the density map, and a density map with the same contour level, but colored by domain/protein is now shown in Revised Fig. 2.

Based on the cryo-EM maps of the BRAF:14-3-3₂ and BRAF:14-3-3₂:MEK complexes, careful model building and refinement was performed to construct the BRAF:14-3-3₂ and BRAF:14-3-3₂:MEK structures, revealing that the overall C α RMSD for the corresponding BRAF:14-3-3₂ portions of the two structures was 0.95 Å and that the C α RMSD of the BRAF KDs was 0.99 Å (with the largest difference of 3.2 Å at residues 472-482). In the MEK-free structure, we can clearly see the C α chain position of the critical elements that define the inactive KD conformation: the α C-helix was in the “out” conformation, residues A598-S602 formed the inhibitory turn, the R spine was broken (with residue L505 misaligned), and the closer orientation of the N- and C-lobes (as reported by Park *et al.*, 2019 and Liao *et al.*, 2020) was observed. Our revised manuscript now includes data presented in Supplementary Table 2 showing a comparison of the above-mentioned structural elements fitted to the cryo-EM density maps of BRAF:14-3-3₂ and BRAF:14-3-3₂:MEK. Representative density regions in the KD domain C-lobe are also presented to further demonstrate the quality of the density fit. In addition, our revised manuscript now includes data presented in Supplementary Table 3, showing the fitting of the active (PDB ID 2FB8) and inactive (PDB ID 6U2G) BRAF KD conformations into our BRAF:14-3-3₂ (EMD-23814) and BRAF:14-3-3₂:MEK (EMD-23813) density maps (see response to point 4 below), which confirmed that the BRAF:14-3-3₂ and BRAF:14-3-3₂:MEK reconstructions in this study represent the inactive conformation. Finally, we have modified the text of the revised manuscript to be more nuanced in our conclusions and to state that MEK binding may further stabilize the autoinhibited state.

(c) The authors propose RBD interactions with 14-3-3 maintain the inactive BRAF complex, and that Ras binding to the RBD dislodges the RBD from 14-3-3 given steric clashes. This is proposed as the first step toward BRAF activation. Regarding RBD-14-3-3 interactions, given the middling effect in the BRAFREG:BRAF KD BRET assay of point mutations to the RBD compared to the T241P point mutation in the CRD, seeing the effect of a more stringent alteration (e.g. mutation to negatively charged residues or deletion of the RBD), would help to better assess the importance of this interaction. The “dislodging” activity of Ras remains speculative (Ras binds tightly to both BRAF:(14-3-3)₂ complexes and the authors do not test altering residues contributing to the proposed Ras:RBD clash), but proving this model could constitute a new study beyond the scope of this manuscript.

RESPONSE:

We think the reviewer may have misinterpreted our view regarding RAF autoinhibition and the origin of the steric clashes in our model. First, the steric clashes caused by contact between RAS and the autoinhibited BRAF structure occur between RAS and 14-3-3, not RAS and the RBD as indicated in the reviewer’s comment. In the autoinhibited structure, the four basic RBD residues that form critical ionic

bonds with RAS are largely exposed, but formation of these bonds would generate steric clashes and electrostatic repulsion between RAS and 14-3-3. Second, we do not propose that “RBD interactions with 14-3-3 maintain the inactive BRAF complex”, only that they contribute to BRAF autoinhibitory interactions. It has long been known that the CRD plays a key role in RAF autoinhibition. The study by Park *et al.*, 2019, further demonstrated the importance of the CRD by providing its exact placement and contacts with the BRAF KD and 14-3-3, which are critical for maintaining the autoinhibited conformation. Nonetheless, while the CRD plays the predominant role in RAF autoinhibition, we find that the RBD also contributes through its contact with 14-3-3. We think that the RBD interactions are additive, and in conjunction with the critical contacts made by the CRD, contributes to the maintenance of the autoinhibited state and may function to orient the RBD for RAS binding. Because the role of the RBD in RAF autoinhibition is more minor in comparison to the CRD, mutations in the RBD:14-3-3 interface would not be expected to have as large an effect on autoinhibition as did the T241P CRD mutation, which was the case, when the M186A/M187A and M186W/M187W mutants were analyzed in the NanoBRET autoinhibition assay. As suggested by the reviewer, these methionines were mutated to glutamic acid residues to determine if a more stringent alteration would have a larger effect in the NanoBRET RAF autoinhibition assay. Surprisingly, we found that in the RAF autoinhibition assay, the MM>EE mutation resulted in a reproducible ~10% increase in the BRET signal, which may reflect interactions with R222 in α 9-helix of 14-3-3. Moreover, as was observed for the MM>AA mutant, the MM>EE mutant was more effective at suppressing MEK activation mediated by the isolated BRAF-KD than was WT-BRAF-REG. In contrast, the same MM>EE mutations resulted in a near loss in RAS binding, likely due to electrostatic repulsion with E31 and D33 of RAS. Data analyzing the BRAF M186E/M187E mutant are now presented in Revised Fig. 3 and 5.

We would also like to note that in the introduction of our original manuscript, we did affirm that interactions of both the CRD and the 14-3-3 dimer are critical for the autoinhibited state (Page.3 Lines 17-24). Moreover, in the results section describing the NanoBRET RAF autoinhibition assay, we stated: “Because the M186 and M187 residues have the potential to make numerous contacts with 14-3-3 at the interface and given that contacts between 14-3-3 and the CRD are known to play a key role in maintaining the autoinhibited state, we next took a mutational approach to determine whether these RBD residues also contribute to RAF autoinhibition.” Then, we concluded this section by stating “Thus, while the CRD plays the predominant role in autoinhibition, these findings indicate that M186/M187 and the RBD also contribute to the maintenance of the autoinhibited state.”

(d) How do the authors compare their results with the Eck structure, where the RBD was found to be disordered, whereas it is ordered in this manuscript? The difference mentioned is that the sample in this case was from mammalian cells. Was this due to potential post-translational modifications? If so, what are they? Knowing that would be very important to rationalize. Was this due to differences in data collection or data processing OR a real biological difference? What is particular about the mammalian cell expression “more closely represents” authentic signaling complexes?

RESPONSE:

As the reviewer stated, the RBD region in our autoinhibited BRAF complexes was resolved, whereas the RBD was disordered in the structure reported by Park *et al.*, 2019, which was the first structural report of this important complex. Park and coworkers (2019) used an insect cell expression system to isolate their BRAF complexes, whereas our complexes were obtained from human 293FT cells stably expressing Halo-BRAF. It is possible that differences in the post-translational modification of components in the BRAF complexes could be a contributing factor; however, to determine with certainty how our complexes differ from the complexes obtained by Park and coworkers, a detailed qualitative and

quantitate analysis of the post-translational modifications occurring on individual components of the complexes would need to be conducted in both systems, in parallel, and such an analysis is beyond the scope of this study. In the absence of a thorough side-by-side comparison of the complexes, we respectfully decline to speculate further on why the RBD was resolved in our complexes, but not in those isolated by Park *et al.*, 2019.

To answer the reviewer's question regarding whether the cryo-EM sample preparation, data collection, or processing may have contributed to the different outcome between the two studies, we can state that cryo-EM specimen preparation was key to overcoming the problem of "preferred particle orientation". Intensive optimization tests related to cryo-EM buffer conditions and sample preparation were performed to successfully tackle this challenge. Interestingly, Park *et al.*, 2019 may have encountered a similar problem of "preferred particle orientation", as their methods indicate that they collected data under conditions of stage tilting to increase the number of observed orientations.

Specific comments:

(1) Page 5, line 24 and and Figure S1D: "...the three sample complexes revealed that 14-3-3 ϵ and 14-3-3 ζ constituted approximately 50% and 25%, respectively of the 14-3-3 proteins... all other 14-3-3 isoforms were detected.

Can the authors comment on whether the % of 14-3-3 isoforms in the complexes reflect whole cell % of isoforms? Or is BRAF preferentially forming complexes with 14-3-3 ϵ and 14-3-3 ζ ? If it is, are there any conclusions that one can draw from the structures and interacting residues as to why this may be?

RESPONSE:

Studies analyzing the abundance of 14-3-3 isoforms in various human cell lines and tissues indicate that 14-3-3 ϵ is the most abundant isoform in most human cell types (Wang *et al.*, 2015). In addition, specific analysis of HEK 293 cells indicates that 14-3-3 ϵ and 14-3-3 ζ are the two most abundant 14-3-3 isoforms in these cells (Geiger *et al.*, 2012 and Gogl *et al.*, 2021). From our semi-quantitative analysis, we find that the 14-3-3 ϵ represents ~43-45% (most abundant) of the 14-3-3 proteins in these cells, whereas 14-3-3 ζ represents ~17% (second most abundant). Studies have also found that 14-3-3 ϵ preferentially heterodimerizes, while other isoforms tend to indifferently homodimerize or heterodimerize (Yang *et al.*, 2006). Therefore, the finding that 14-3-3 ϵ constituted approximately 50% of the 14-3-3 protein in our complexes is consistent with 14-3-3 ϵ being the most abundant isoform in these cells and being the isoform that preferentially heterodimerizes. The identification of 14-3-3 ζ as the next most abundant 14-3-3 protein in our complexes is also consistent with 14-3-3 ζ being the second most highly expressed 14-3-3 isoform in these cells. In the revised manuscript, we now state that "Mass spectrometry analysis of each of the three sample complexes revealed that 14-3-3 ϵ and 14-3-3 ζ constituted approximately 50% and 25%, respectively of the 14-3-3 proteins present in both the monomeric and dimeric BRAF complexes (Supplementary Fig. 1d), a finding consistent with the expression level of these 14-3-3 isoforms in 293FT cells and the propensity for 14-3-3 ϵ to heterodimerize."

(2) Page 6, line 15: "despite the presence of MEK in the cryo-EM sample fraction, no density corresponding to MEK was observed in the 3D reconstructions..." and Figure S1A-C.

Can the authors quantify relative band densities of BRAF, 14-3-3 and MEK in EM sample fractions in Fig. S1A-C? MEK appears sub-stoichiometric in all samples, but particularly reduced in the BRAF2:14-3-32 complex.

RESPONSE:

As suggested by the reviewer, we have quantified the relative band densities from the gel fractions used for cryo-EM analysis, and this information has been added to figure legend of Supplementary Fig. 1. However, it is important to note that these are silver stained gels, and because the interaction of the silver ions is strongest with certain functional groups in specific amino acids (Asp, Glu, His, Cys, and Lys) and the content of these functional groups can vary among proteins, the relative band densities are not necessarily proportional to their stoichiometric ratio. In particular, MEK contains fewer of the above-mentioned functional groups when compared with BRAF (68 for MEK versus 86 for BRAF), which likely contributes to the lower band intensity in the silver-stained gels.

Could the lack of a MEK inhibitor in the sample also explain missing MEK density when compared to results from Park et al (PDBID 6q0j) or Haling et al (PDBID 4mne)?

RESPONSE:

We are assuming the reviewer is referring to the lack of MEK density in the BRAF₂:14-3-3₂ complex, given that MEK was present in fraction 16, but no density for MEK was observed in the cryo-EM density maps. Our dimeric complexes were isolated under conditions where the BRAF dimers were stabilized by treating cells with the type I RAF inhibitor SB590885. We did not try co-treating cells with both the RAF inhibitor and a MEK inhibitor. It is possible that cotreatment using a MEK inhibitor that stabilizes the BRAF:MEK interaction, such as CH5126766, may have resulted in dimeric BRAF complexes and density maps that contained MEK.

In addition to cellular BRAF complexes not containing MEK, it also seems likely MEK could be dissociating from intact MEK:BRAF:14-3-3 complexes during the purification process given reported BRAF:MEK affinities (50-100 nM), whereas 14-3-3 stays bound due to avidity considerations (see comments above).

RESPONSE:

As stated above in our response to point (a), we do not think that MEK was simply dissociating from our monomeric BRAF complexes during the purification process. In our studies, we isolated BRAF monomeric complexes from serum-depleted cells that were either treated with the MEK inhibitor CH5126766 or not, and under **both** conditions, we consistently obtained a proportion of monomeric BRAF:14-3-3₂ complexes (fraction 19) that was equivalent to the monomeric BRAF:14-3-3₂:MEK complexes (fraction 18, See Supplementary Fig. 1a, b). If MEK was simply dissociating, one would expect that there would be an increase in the proportion of "MEK-containing complexes (fraction 18)", when the complexes are isolated from cells treated with CH5126766, which stabilizes the MEK:BRAF interaction. In addition, the purification of our autoinhibited BRAF complexes was quite straightforward and involved affinity capture of the complexes from cell lysates, cleavage of the complexes from affinity resin, and separation of the samples via gel chromatography, which was achieved in approximately 5 hrs, with the samples being maintained at 4°C. Therefore, we think that the isolation of two monomeric BRAF complexes is indicative of distinct cellular pools of autoinhibited BRAF complexes, one that contains MEK and one that is MEK free. Moreover, it is well established that the translation and processing of the RAF kinases in complex, involving protein chaperones such as Hsp90/cdc37 and changes in phosphorylation state. It is just as possible that the MEK-free monomeric state could be a precursor to the MEK-bound state and that MEK binding further stabilizes the autoinhibited conformation.

(3) Page 6, line 26: "our structure superimposing well onto the ligand-free dimer structure, with an

overall C α root mean square deviation (RMSD) of 1.03 Å”

pg. 7, line 18 “overlap well with the high resolution crystal structure of inhibitor (GDC-0879)-bound BRAF KD (residues D432–R735)₂:14-3-3₂ complexes (C α RMSD of 0.80 Å)” and Figure 1C.

Is the RMSD specific to the BRAFKD dimer, or the overall BRAF₂:14-3-3₂ complex?

RESPONSE:

The C α RMSDs values of 1.03 and 0.8 Å are for the BRAF protomer used to perform the alignments of the BRAF₂:14-3-3₂ complexes. In the revised text, we now state that the overall alignment of our BRAF₂:14-3-3₂ dimer with dimers PDBID 6UAN (Kondo *et al.*, 2019) and PDBID 6U2H (Liau *et al.*, 2020) yielded C α RMSDs values of 1.54 and 2.29 Å, respectively.

In Fig 1C it would be useful to align the BRAF₂:14-3-3₂ complex reported in this manuscript against the Kondo et al structure (6uan) using only the 14-3-3 dimer or only the BRAF dimer. It appears the pose of the BRAF dimer relative to 14-3-3 differs between those two structures? To that end, it would be useful to align the current structure against the much higher resolution Liau et al crystal structure (6u2h or 6xag). Do the two EM structures and the crystal system represent different juxtapositions of 14-3-3 dimer vs. BRAF dimer? In Fig 1C, can the authors more clearly label the grey structure as 6UAN vs. current structure? 6UAN is difficult to make out with current coloring.

RESPONSE:

After a systematical alignment of our BRAF dimer structure with the dimer structures of PDBID 6UAN (Kondo *et al.*, 2019) and PDBID 6U2H (Liau *et al.*, 2020), using either an individual BRAF protomer, the BRAF dimer, or the 14-3-3 dimer, we find that except for small local differences, the overall position of the BRAF dimer relative to the 14-3-3 dimer is similar among the three structures. However, the largest difference observed is that one 14-3-3 protomer in the Liau crystal structure is tilted closer and contacts one of the BRAF kinase domains, which Liau *et al.*, 2020 indicated in their paper “is probably a result of the constraints of the crystallographic lattice.” As the reviewer suggested, we have now included in Supplementary Fig. 2f an overlay of our structure with PDBID 6U2H (Liau *et al.*, 2020). In addition, the coloring of 6UAN in Fig. 1C has been changed to a darker gray to improve visibility, per the reviewer’s suggestion.

(4) Page 8, line 4: “Strikingly, except for the presence or absence of MEK, the overall conformation of the monomeric BRAF structures was nearly identical...”

line 11 “...they demonstrate that MEK binding is not required for BRAF to assume a stable, autoinhibited conformation. The BRAF kinase domain in both our monomer structures displayed the canonical kinase inactive conformation...”

line 17 “the structure of the KDs displayed an almost identical conformation as did recent inactive KD structures bound to ATP analogs...”

line 23 “These findings indicate that the ATP binding pocket of the BRAF KD is stable in the “apo” nucleotide-free state”.

P. 14, line 22 “our structures show that the ATP-binding site is fully formed in the “apo” state and that stabilization of the autoinhibited state is also not dependent on ATP binding” Fig. 2C, and Fig S3B

The authors comment multiple times that except for MEK occupancy the BRAF:(14-3-3)₂:MEK and BRAF:(14-3-3)₂ structures are identical, including secondary structure details of the BRAF ATP-binding pocket. Figure 2A only shows the BRAF:(14-3-3)₂:MEK map, implying a similar map for BRAF:(14-3-3)₂. However, the map presented for BRAF:(14-3-3)₂ in Fig. S3B shows significantly poor density for BRAFKD,

and the N-lobe in particular looks to be highly flexible/poorly resolved and is likely due to the N-lobe taking on multiple conformations. This suggest MEK and ATP do influence BRAF conformation in the inactive complex and this map ought to be included in the main text figures

RESPONSE:

As indicated in point (b) above, our revised manuscript now contains Supplementary Table 2, which shows a side-by-side comparison of the critical density regions from the KDs of the BRAF:14-3-3₂ and BRAF:14-3-3₂:MEK complexes. The C α main chain is resolved in the KD N-lobe of the BRAF:14-3-3₂ structure. We agree that the kinase domain of the BRAF:14-3-3₂ complex, may be more flexible than that of the BRAF:14-3-3₂:MEK complex; however, we can clearly discern that the kinase domain is in the inactive conformation with the α C- helix in the “out” position, residues A598-S602 forming the inhibitory turn, the R spine misaligned, and the N- and C- lobes of the KD in a closer configuration than is observed in RAF inhibitor-bound inactive structures (as reported by Park *et al.*, 2019 and Liao *et al.*, 2020).

In addition, as requested by the reviewer, we have added the cryo-EM map of BRAF:14-3-3₂ to Fig. 2 and we are more nuanced in our comments and state that “our structures suggest that binding of the 14-3-3 dimer and RAF intramolecular interactions are sufficient to maintain the autoinhibited conformation, but that MEK binding may further stabilize the complex, as evidenced in the local resolution maps (Supplementary Fig. 3a-b).”

Can the authors show both a BRAFKD protomer in the active (i.e. PDB ID 2fb8, 4mnf or similar) and inactive conformation (i.e. 6pp9 or 6u2g) fit into density for the BRAF-C lobe in the BRAF:(14-3-3)₂ map to better assess the conformation taken on by the BRAF-N-lobe in this map?

RESPONSE:

In accordance with the reviewer’s request, we now include as Supplementary Table 3, the fitting of an active (PDB ID 2FB8) and an inactive (PDB ID 6U2G) BRAF KD conformation into the BRAF KD density maps of BRAF:14-3-3₂ (EMD-23814) and BRAF:14-3-3₂:MEK (EMD-23813), which confirms that the BRAF kinase domains in our BRAF:14-3-3₂ and BRAF:14-3-3₂:MEK complexes represent the inactive conformation.

Can the authors show and compare local EM density between both +/- MEK structures for the BRAF and MEK ATP binding pocket, including key elements such as the P-loop (res 463-471 in BRAF) and inhibitory turn (res 598-602) to support claims that ATP binding has no effect on this conformation for either structure?

RESPONSE:

As suggested by the reviewer, now included in Supplementary Table 2 and shown below are the cryo-EM densities of the active site regions in the BRAF:14-3-3₂ and BRAF:14-3-3₂:MEK maps. Although our analysis does not allow us to determine complete side chain details in the active site, the main C α chain position is discernable and indicates that the ATP binding pocket is formed in our BRAF:14-3-3₂ and BRAF:14-3-3₂:MEK structures.

(5) p. 9, line 17: “A distinct feature of the BRAF structures reported here is the resolution of the RBD in the context of the full-length, autoinhibited BRAF monomer” line 23: “an extensive contact surface of ~435 Å² is observed between the RBD and the 14-3-3 protomer”, Fig 2C:

Can the authors comment/speculate as to why the RBD and this interface were not resolved in Park et al structure, but have clear density in the maps reported here?

RESPONSE:

Please see our response to point (d) above, where this comment was addressed.

Are the key residues conserved in insect 14-3-3?

RESPONSE:

Yes, the amino acid sequence of the 14-3-3 α 8- and α 9- helices is highly conserved among the different human isoforms and with insect 14-3-3 proteins. Further, key α 8- and α 9-helix residues at the RBD:14-3-3 interface are fully conserved between the human and insect proteins.

Can the authors overlay semi-transparent map density in Fig 2C to help the reader evaluate this interface? Is there side-chain density for M186 and M187, given that the overall resolution of the reconstruction appears to be best around this RBD/1433 region?

RESPONSE:

We thank the reviewer for this suggestion and now included as Supplementary Fig. 6B (shown below) a figure showing the overlay of a semi-transparent density map onto the RBD:14-3-3 interface. In addition, the density map for the entire RBD, with the M186 and M187 residues labeled (shown below), has been added to Supplementary Fig. 4. As indicated in the maps, we do see some side chain density for these residues.

(6) p. 10, line 21 “a reproducible 10% decrease in signal was observed for the M186W/M187W mutant, whereas a 20% increase in signal was observed for the M186A/M187A mutant” Fig 2D: Do the authors believe a 10% reduction in BRET signal represents the full effect of ablating the RBD/14-3-3 interface, as compared to 50% reduction seen for the T241P CRD mutant? Given subtle effects of M186/M187 mutants, it would be helpful to see the effect of harsher mutations (e.g. to negatively charged residues) or deletion of the RBD in the BRET assay to confirm the full effect of breaking the RBD/14-3-3 on the BRAF inactive complex.

RESPONSE:

Please see our response to point (c) above, where this comment was addressed.

(7) p. 11, line 19 “both complexes exhibited a high degree of binding and had affinities in the nanomolar range.” and Fig. 3C. Can the authors comment on how the measured Ras affinities to the BRAF:14-3-3:MEK and BRAF:14-3-3 complexes compare to Ras binding to the BRAF-RBD alone? Does presence of the full complex increase, decrease or have no effect on Ras binding? On that end, Does the RBD bind to 1433 directly with reasonable affinity (even if it is micromolar?) – This could be tested directly.

RESPONSE:

The binding affinity of our full-length, autoinhibited BRAF complexes to KRAS was in the nanomolar range (85-127 nM), which is comparable with the binding affinities reported for the isolated RAF RBD to KRAS (~50-200 nM, depending on the binding assay used and the ionic strength of the buffer) (Fetics *et al.*, 2015; Hobbs *et al.*, 2020; and Tran *et al.*, 2021).

We have been unable to measure a binding affinity between the isolated BRAF-RBD and 14-3-3. However, we do not think this finding precludes that interactions between the RBD and 14-3-3 do occur at the RBD:14-3-3 interface in the context of high affinity 14-3-3 binding to the full-length, autoinhibited BRAF complexes, which is mediated by the pS365 and pS729 sites. It is not uncommon for secondary protein:protein interactions to occur only in the context of another high affinity interaction.

(8) p. 12 line 7: “Specifically, RAS residues I21, Q22 (α 1-helix), Q25-E31 (SI region), K42 and V45 (β 2 sheet) would clash with 14-3-3... this region of 14-3-3 has a notable negative charge and, while complementary with the positively charged RBD interface, would cause electrostatic repulsion with

negatively charged residues in KRAS switch I... residues D30 and E31 would be brought in close proximity to D197 and E198 of 14-3-3.” This structural analysis predicts point mutants to Ras relieving steric clashes and electrostatic repulsion would improve Ras binding to inactive RAF complexes. If proved true, this would better support speculation that Ras acts in part by dislodging the RBD from 14-3-3.

RESPONSE:

In the autoinhibited conformation, the BRAF RBD is oriented towards the “RAS pocket” with the critical ionic bond forming residues exposed. In this position, one might think that steric clashes and electrostatic repulsion between 14-3-3 and RAS might impede RAS contact and be inhibitory to the RAS-RAF interaction. However, because some of the RBD residues that contact RAS are occluded by 14-3-3, we believe that the steric clashes and electrostatic repulsion represent a critical step in the monomer to dimer transition and RAF activation, as it would be the initial event that instigates a change in how the 14-3-3 dimer contacts BRAF, which would allow the occluded RBD residues to be exposed, and ultimately lead to the dislodging of the RBD and CRD, release of 14-3-3 from the S365 site, and disruption of the autoinhibited state. Point mutations in RAS that might be expected to relieve the steric clashes/electrostatic repulsion could fail to expose the full RAS binding interface of the RBD. Moreover, given that the RAS residues implicated are completely conserved, with some found in the critical switch I region required for effector interactions, we feel that mutational analysis of these residues would require an in-depth investigation, evaluating the effects of the mutations on RAS structure, GTP binding, as well as interactions with various effectors, including RAF. We agree that additional experimentation is needed to prove our model; however, as indicated in the reviewer’s comment in point (c), such a study would constitute a new project and be beyond the scope of the current manuscript.

(9) Minor points:

- Page 5 – SB 590885 is a Type 1 inhibitor;

RESPONSE:

We now state “the Type 1 BRAF inhibitor SB590885” when we first introduce this inhibitor.

- Figure 2F – It is unclear if the protein expression levels are equal for the bottom two plates. Please control for this as the cause of the differences in colony counts;

RESPONSE:

Protein levels are now included for the focus forming assay (Revised Fig. 3e).

- Figure 4D: It may be better to show all the figures with the same perspective. It is difficult to compare the four panels when the model moves only slightly between each of them.

RESPONSE:

We appreciate the suggestion, and this figure has been modified accordingly (now Revised Fig. 5e).

Finally, the title is suggestive of more than what is presented in the manuscript. The monomer to dimer transition for RAF involves much more than Ras binding. The SHOC2-pp1c-Ras complex needs to dephosphorylate the pS-365 (BRAF), two RAF molecules need to come together and one 1433 dimer molecule needs to be displaced and set free. How all of that occurs is yet to be discovered. It is possible to conceive other intermediate steps. So, the title could be modified to better reflect the content of the manuscript.

RESPONSE:

We fully agree that the RAF activation process is highly complex and involves regulatory proteins that promote a change in RAF subcellular localization, changes in phosphorylation state of RAF, as well as changes in RAF protein interactions, which ultimately results in the transition from autoinhibited monomers to active BRAF dimers. Our title is not attempting to describe the entire RAF activation process, but we think it does provide structural insights as to how RAS binding can initiate this process and the structure changes that must occur for BRAF to transition from a monomer to a dimer.

Overall, the manuscript contains data that could be considered for publication, given the authors address the concerns / comments listed above.

Reviewer #3 (Remarks to the Author):

In this manuscript by Martinez Fiesco, Durrant and colleagues, the group reports on several structures of BRAF isolated from mammalian cells. This includes cryoEM structures of a BRAF dimer bound to 14-3-3 isolated in complex with the inhibitor SB590885, as well as structures of the BRAF monomer bound to 14-3-3 but in the presence and absence of MEK. The structures show similar features to recently reported cryoEM structures reported by Park et al., however, the authors in this manuscript also reveal the position of the RAS-binding domain (RBD), which was not resolved or previously reported in other structures. The insights on the RBD and 14-3-3 interactions are highly novel and allow the authors to propose a model for how the monomer to dimer transition in BRAF is mediated by RAS binding. The manuscript is well written and the implications of the structures to our understanding of RAS-RAF signaling are supported through mutational and functional studies, making this a highly compelling body of work. I have listed a few points below for the authors - the request of additional mutations within the RBD could be considered but are not essential.

RESPONSE:

We thank the reviewer for the positive comments.

Specific comments.

1. Can the authors mutate the interface residues M186/M187 to negatively charged amino acids (eg. MM>EE)? It would be interesting to see how such mutations compare in the assays shown in Figure 2D-F. Based on their model, such mutations may disfavor binding to 14-3-3 but also RAS binding.

RESPONSE:

We greatly appreciate the reviewer's suggestion. We have generated the MM>EE mutant and have found that these mutations do not disrupt BRAF autoinhibitory interactions but rather result in a reproducible 10% increase in the BRET signal in the NanoBRET autoinhibition assay, which may reflect potential interactions with R222 in the α 9-helix of 14-3-3. Strikingly, these mutations severely disrupt RAS binding in all assays tested (BRET, co-immunoprecipitation, and pull-down assays). Modeling of the M186E/M187E substitutions indicates that there would be significant charge repulsion due to the presence of KRAS residues E31 and D33. Analysis of the M186E/M187E mutant is now included in Revised Fig. 3 and 5.

2. Can the authors add additional panels or images to clarify Figure 3F? The arguments and novelty in this section are potentially very high but the images could be improved to better illustrate specifically the repulsion between RAS and 14-3-3 that could facilitate rearrangement of BRAF to dislodge the auto-inhibited complex.

RESPONSE:

We thank the reviewer for this suggestion. We have revised this figure (now Revised Fig. 4f) and have included additional panels, which we think helps to depict the repulsion between RAS and 14-3-3.

3. Part of the alignment in Figure 2C appears to be cut-off.

RESPONSE:

The full sequence alignment is now shown in Revised Fig. 3B.

4. In Figure S3D, can the authors add images of the entire RBD showing similar model and map overlays; several orientations can be shown. Such images would be complementary to the individual segments already shown in this panel but would give a better sense of map quality and fit throughout this domain.

RESPONSE:

We thank the reviewer for this suggestion. Now included in Supplementary Fig. 4 are images showing structure and density map overlays of the entire RBD.

References

Bery, N. *et al.* BRET-based RAS biosensors that show a novel small molecule is an inhibitor of RAS-effector protein-protein interactions. *eLife* **7**, e37122, doi:10.7554/eLife.37122 (2018).

Fetics, S. K. *et al.* Allosteric effects of the oncogenic RasQ61L mutant on Raf-RBD. *Structure* **23**, 505-516, doi:10.1016/j.str.2014.12.017 (2015).

Geifer, T. *et al.* Comparative proteomic analysis of eleven common cell lines reveals ubiquitous but varying expression of most proteins. *Mol Cell Proteomics* **11**, M111.014050, doi:10.1074/mcp.M111.014050 (2012).

Golg, G. *et al.* Hierarchized phosphotarget binding by the seven human 14-3-3 isoforms. *Nat Commun* **12**, 1677, doi:10.1038/s41467-021-21908-8 (2021).

Hobbs, G.A. *et al.* Atypical KRAS^{G12R} Mutant Is Impaired in PI3K Signaling and Macropinocytosis in Pancreatic Cancer. *Cancer Discov* **10**, 104-123, doi: 10.1158/2159-8290.CD-19-1006 (2020).

Kondo, Y. *et al.* Cryo-EM structure of a dimeric B-Raf:14-3-3 complex reveals asymmetry in the active sites of B-Raf kinases. *Science* **366**, 109-115, doi:10.1126/science.aay0543 (2019).

Jin, T. *et al.* RAF inhibitors promote RAS-RAF interaction by allosterically disrupting RAF autoinhibition. *Nat Commun* **8**, 1211, doi: 10.1038/s41467-017-01274-0 (2017).

Lavoie, H. *et al.* Inhibitors that stabilize a closed RAF kinase domain conformation induce dimerization. *Nat Chem Biol* **9**, 428-436, doi:10.1038/nchembio.1257 (2013).

Park, E. *et al.* Architecture of autoinhibited and active BRAF-MEK1-14-3-3 complexes. *Nature*, doi:10.1038/s41586-019-1660-y (2019).

Tran, T. H. *et al.* KRAS interaction with RAF1 RAS-binding domain and cysteine-rich domain provides insights into RAS-mediated RAF activation. *Nat Commun* **12**, 1176, doi:10.1038/s41467-021-21422-x (2021).

Wang, M. *et al.* Protein abundance data, integrated across model organisms, tissues, and cell-lines. *Proteomics* **15**, 3163-3168, doi:10.1002/pmic.201400441 (2015).

Yang, X. *et al.* Structural basis for protein-protein interactions in the 14-3-3 protein family. *Proc Natl Acad Sci USA* **103**, 17237-17242. doi:10.1073/pnas.0605779103 (2006).

REVIEWERS' COMMENTS

Reviewer #1 (Remarks to the Author):

The authors addressed all of the concerns and I recommend the paper to be accepted for publishing.

Reviewer #2 (Remarks to the Author):

The revised manuscript is substantially improved compared to the initially submitted version in content as well as the writing. The authors have addressed almost all of my concerns / suggestions. The current work is a meaningful and significant advance in understanding the RAF activation process and I fully support the publication of the manuscript in its current form in Nature Communications.

Reviewer #3 (Remarks to the Author):

The authors have done well with their rebuttals, I have no further concerns.